# Saturation genome editing of 11 codons and exon 13 of *BRCA2* coupled with chemotherapeutic drug response accurately determines pathogenicity of variants

Sounak Sahu[1], Teresa L. Sullivan[1], Alexander Y. Mitrophanov[2], Mélissa Galloux[3], Darryl Nousome[4], Eileen Southon[1], Dylan Caylor[1], Arun Prakash Mishra[1], Christine N. Evans[5], Michelle E. Clapp[5], Sandra Burkett[1], Tyler Malys[2], Raj Chari[5], Kajal Biswas[1¤], Shyam K. Sharan[1]*

1 Mouse Cancer Genetics Program, Center for Cancer Research, National Cancer Institute, Frederick, Maryland, United States of America, 2 Statistical Consulting and Scientific Programming, Frederick National Laboratory for Cancer Research, National Institutes of Health, Frederick, Maryland, United States of America, 3 Independent bioinformatician, Marseille, France, 4 CCR Bioinformatics Resource, Leidos Biomedical Sciences, Inc. Frederick National Laboratory for Cancer Research, Frederick, Maryland, United States of America, 5 Genome Modification Core, Laboratory Animal Sciences Program, Frederick National Laboratory for Cancer Research, Frederick, Maryland, United States of America

¤ Current address: Division of Cancer Prevention, National Cancer Institute, Bethesda, Maryland, United States of America

* sharans@mail.nih.gov

**Data Availability Statement:** All data generated or analyzed during this study are included in the manuscript and in the supplementary files. The raw

## Abstract

The unknown pathogenicity of a significant number of variants found in cancer-related genes is attributed to limited epidemiological data, resulting in their classification as variant of uncertain significance (VUS). To date, *Breast Cancer gene-2 (BRCA2)* has the highest number of VUSs, which has necessitated the development of several robust functional assays to determine their functional significance. Here we report the use of a humanized-mouse embryonic stem cell (mESC) line expressing a single copy of the human *BRCA2* for a CRISPR-Cas9-based high-throughput functional assay. As a proof-of-principle, we have saturated 11 codons encoded by *BRCA2* exons 3, 18, 19 and all possible single-nucleotide variants in exon 13 and multiplexed these variants for their functional categorization. Specifically, we used a pool of 180-mer single-stranded donor DNA to generate all possible combination of variants. Using a high throughput sequencing-based approach, we show a significant drop in the frequency of non-functional variants, whereas functional variants are enriched in the pool of the cells. We further demonstrate the response of these variants to the DNA-damaging agents, cisplatin and olaparib, allowing us to use cellular survival and drug response as parameters for variant classification. Using this approach, we have categorized 599 *BRCA2* variants including 93-single nucleotide variants (SNVs) across the 11 codons, of which 28 are reported in ClinVar. We also functionally categorized 252 SNVs from exon 13 into 188 functional and 60 non-functional variants, demonstrating that saturation genome editing (SGE) coupled with drug sensitivity assays can enhance functional annotation of *BRCA2* VUS.

sequencing reads are deposited to GEO database and can be accessed from https://www.ncbi.nlm.nih.gov/geo/query/acc.cgi?acc=GSE238143 (Accession number: GSE238143). The code developed for data analysis is available on Github: https://github.com/dnousome/CRISPRAnnotation

**Funding:** This work was supported by the Intramural Research Program, Center for Cancer Research, National Cancer Institute, U.S. National Institutes of Health (SKS) and using federal funds from the National Cancer Institute, National Institutes of Health (under contract No. HHSN261201500003I to RC). The funders had no role in study design, data collection and analysis, decision to publish, or preparation of the manuscript.

**Competing interests:** The authors have declared that no competing interests exist.

## Author summary

The exponential rise in genetic sequencing led to the identification of several missense variants. However, its clinical utility is often limited by our ability to determine the functional impact of genetic variants. This has resulted in the identification of many variant of uncertain significance (VUS), particularly in *BRCA2*. Several functional assays have been developed to ascertain the impact of VUSs on protein function that can be used to determine their pathogenicity. Our mouse ES cell (mESC)-based method includes generation of individual *BRCA2* variants in a bacterial artificial chromosome (BAC) by recombineering and assess their expression in mouse ES cells. These cells can then be used to determine how the variants affect mESC viability and sensitivity to DNA damaging agents. Although the assay has high sensitivity and specificity, the process is time consuming and labor-intensive. To overcome these limitations, here we report the development of a CRISPR-Cas9 based high-throughput approach that can be used to classify multiple *BRCA2* variants. Specifically, we utilized the BRCA2-independent single-stranded annealing pathway to efficiently knock-in targeted mutations in mESCs using a library of single-stranded DNA donors. Not only do our results support the ClinVar classification of some variants, but we also assign clinical significance to several VUSs, for which no functional data is currently available.

## Introduction

Clinical interpretation of genetic variants is still a bottleneck for personalized medicine. Only 2% of missense variants have a clinical classification, which imposes significant challenges for patients and clinicians for genetic counseling and risk assessment [1]. The American College of Medical Genetics and Genomics (ACMG) has recommended standards and guidelines for genetic variant classification based on population, computational, functional and segregation data. Currently, a pathogenic variant can be distinguished from a benign variant based on its co-segregation with the disease within families, as well as its frequency in the control population data [2]. Many of these variants are rare and have limited epidemiological and family linkage data. Consequently, they remain unclassified and are referred to as variants of uncertain clinical significance (VUS) in the ClinVar (https://www.ncbi.nlm.nih.gov/clinvar/), database that contains clinical variant reports [3]. VUSs have been identified in several DNA repair genes, of which *BRCA1* and *BRCA2* are two of the clinically actionable genes that contain the highest number of VUSs [1]. The proteins encoded by these genes play a major role in the repair of DNA double stranded break (DSB) by homologous recombination (HR) [4].

The lifetime risk of developing breast, ovarian and other cancers is significantly increased in *BRCA1* and *BRCA2* mutation carriers [5,6]. Individuals with a personal or family history of early onset and/or bilateral breast and/or ovarian cancer, or a history of male breast cancer, are offered sequencing-based genetic tests to screen for mutations in *BRCA1* and *BRCA2* [7,8]. This has led to the identification of many unique variants in these genes [9–12]. In response to the rapid increase in the number of VUSs, the Global Alliance for Genomics and Health created clinical archives, such as the BRCA Exchange (https://brcaexchange.org/), which contains information on more than 71,800 distinct BRCA mutations [13]. Of these, only 7,445 variants have been classified by the Evidence Based Network for the Interpretation of Germline Mutant Alleles (ENIGMA) expert panel (https://enigmaconsortium.org/). A number of computational algorithms are currently available to predict the impact of a variant on protein structure and

function, but they often provide conflicting interpretations [14,15]. Several multiplex assays of variant effect (MAVE) approaches, computational models and functional assays have been developed to evaluate the functional impact of VUSs [16]. These assays examine the ability of BRCA1 and BRCA2 variants to support cell viability, sensitivity to DNA damaging agents, ability to repair DSBs by HR, or to exhibit transcriptional activity and predict the role of mutations in cancer-predisposition [16–25].

Recently, many CRISPR-based high-throughput screening (HTS) assays have also been reported to classify variants of DNA repair genes [26–30]. HTS involving CRISPR-based base editors quantifies the presence of the single-guide RNA (sgRNAs) and not the nucleotide changes in the genomic DNA [27,28]. While prime editing is a powerful genome-editing technique, the efficiency of prime editors relies on the length of primer-binding site incorporating the desired mutation [29,31]. Hence, each of these programmable editors has its own limitations. Moreover, HR-based saturation genome editing relies on blocking the Non-homologous end-joining (NHEJ) pathway by DNA ligase 4 loss, which has been instrumental in classifying *BRCA1* VUSs [30]. While it enhances variant generation, it may lead to cell lethality, and that can be detrimental to classifying clinically relevant variants of other DNA repair genes. For example, loss of DNA ligase 4 leads to nuclear fragmentation and increased genomic instability in BRCA2-deficient cells [32]. To address these shortcomings, we took an approach utilizing CRISPR-Cas9 based homology-directed repair (HDR) to knock-in DNA sequences without inhibiting other DNA repair pathways. Specifically, we have developed a high-throughput assay to examine the functional significance of BRCA2 variants using single-stranded donor oligodeoxynucleotides (ssODN). Variants are generated by DNA double strand break repair using single-stranded annealing pathway (SSA), which is BRCA2-independent [33,34]. Unlike base editing, we confirm variant generation and quantify them by directly sequencing the edited region using Next Generation Sequencing (NGS) (**S1 Fig**).

As a proof-of-principle, we saturated 11 codons of BRCA2 in exons 3, 18, and 19, leading to the generation and functional categorization of 599 variants representing all possible nucleotide combinations in a particular codon. We selected these specific residues as few of them have been previously studied by multiple laboratories, allowing us to use them as known controls to validate our assay. We selected different exons for the broad applicability of this tool across different exons. We demonstrate that single and multi-nucleotide variants were generated with a strong correlation between replicates. Furthermore, to scale up this SGE approach we also classified all the possible single-nucleotide variants of exon 13. We generated a function score (FS) representing the frequency of variants in the final pool relative to the initial pool. The FSs of all variants were then used in a statistical model to calculate the probability of impact on function (PIF) of the variants based on cell survival and their response to DNA damaging drugs. We anticipate that these results will be useful for understanding the functional consequences of *BRCA2* variants and will be applied to saturate other exons of *BRCA2*.

## Results and discussion

### Generation of a humanized-mouse embryonic stem cell line with a single copy of *BRCA2*

Humanized mESCs-based functional assays have been previously used to distinguish between functional and non-functional variants [20,35–38]. However, the recombineering-based approach of generating variants in a Bacterial Artificial Chromosome (BAC) containing human *BRCA2* is time-consuming and imposes several limitations to multiplex variant generation [35,36]. CRISPR-based genome editing technologies have simplified the generation of desired mutations in mammalian cells. Furthermore, use of mammalian haploid cells has been

immensely informative in the identification of cellular phenotypes. This allows one to perform high-throughput genome editing studies by bypassing the possibility to have an unedited allele or harboring a different mutation in the other allele that can lead to confounding cellular phenotypes. For example, haploid human cells-HAP1 [30,39] or haploidized HEK293T cells [29] or haploidized human pluripotent stem cells [40] have been recently used to perform saturation genome editing to functionally classify a large number of genetic variants. To overcome the shortcomings of our BAC-based approach, and appreciating the advantages of using haploid cells, we have generated a humanized-mESC by integrating a single copy of human *BRCA2* in PL2F7 mESC. These cells contain a functionally null allele (*ko*) and a conditional allele (*cko*) of *Brca2* with flanking *loxP* sites along with two halves of the human *HPRT1* minigene [35] (**Fig 1A**). We tested the presence of the human *BRCA2* in PL2F7 ES cell line by Southern analysis. We identified a clone, PL2F7/F7, with a single copy of the transgene, based on the signal intensity (**Fig 1B**). We confirmed that a single copy of human *BRCA2* transgene was present in this clone by qPCR (**Fig 1C**). We deleted the conditional allele of mouse *Brca2* and confirmed the expression levels of BRCA2 in *Brca2*$^{ko/ko}$ mESCs (**Fig 1D**). Cytogenetic analysis of the PL2F7/F7 ES cells revealed that the human *BRCA2* was integrated in chromosome 12 at position A3 (**Fig 1E**). We further showed that *Brca2*$^{ko/ko}$; *Tg(BRCA2)* cells are proficient in RAD51 foci formation in response to ionizing radiation, suggesting that the *BRCA2* transgene is functional and can substitute the function of the mouse *Brca2* (**Fig 1F**). Additionally, using EdU pulse labelling, we also show normal doubling time in *Brca2*$^{ko/ko}$; *Tg(BRCA2)* ESC clone, suggesting the functional equivalence of humanized mESC to the parental clone Fig 1G.

## Saturation of 11 codons across 3 exons of *BRCA2* variants for functional categorization

Saturation genome editing was performed on PL2F7/F7 mESCs (*Brca2*$^{ko/ko, Tg(BRCA2)}$) using sgRNA and a pool of 180mer ssODNs containing degenerate nucleotides for a particular codon (NNN). Each pool of ssODNs can generate 63 non-wildtype missense variants. We targeted exon 3 and exons 18,19 which encode part of the C-terminal DNA-binding domain of BRCA2 (**S2A Fig** and **S1 File**). The selection of these residues was based on several factors. First, certain residues, such as p.Thr2722, p.Asp2723, have been extensively investigated by numerous laboratories. This widespread study of these residues allowed us to utilize them as established controls, distinguishing between functional or non-functional, thereby serving as reference points for our assay. Second, variants were randomly selected across few exons to ensure the applicability of our approach across various regions of the gene. This selection allowed us to investigate the functional relevance of key domains in BRCA2. Furthermore, sgRNAs were selected based on SpliceAI predictions such that the synonymous PAM mutation does not affect canonical splicing [41]. We employed a SpliceAI score cut off of 0.2 to assess the potential impact of variants on splicing [42].

Degenerate oligos targeting each codon were nucleofected and the cells were pooled together to multiplex variant generation per exon (**Fig 2A**). We generated a total of 599 out of 693 (86.4%) possible combinations of variants corresponding to residues p.Tyr57, p.Glu58 and p.Pro59 of exon 3; p.Thr2722, p.Asp2723, p.Gly2724, p.Cys2765, p.Pro2767 and p.Leu2768 of exon 18; p.Thr2783 and p.Trp2788 of exon 19. The repair kinetics and accuracy of Cas9-induced DSB repair vary based on different factors such as the repair pathway utilized, and the cell type. Recent reports suggest, it takes 10–15 hours to repair majority of Cas9-induced lesions [43,44]. To balance between allowing sufficient time for generation of variants and preventing dropout of non-functional variants, we decided to collect cells at day 3, where we found a good representation of variants.

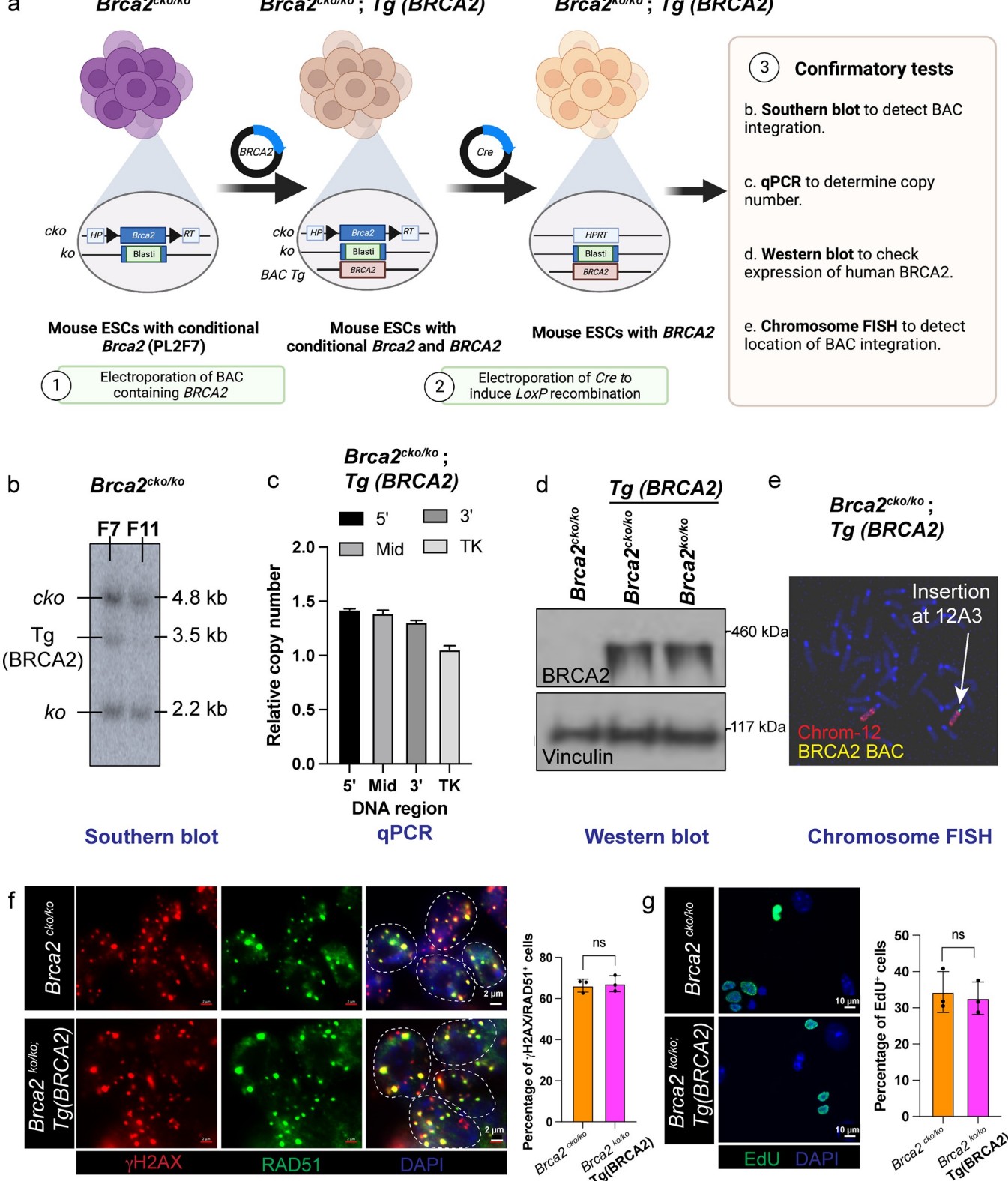

**Fig 1. Human BRCA2 can complement the loss of *Brca2* in mouse embryonic stem cells. (a)** Schematic overview of the generation of mouse ESCs containing a single copy of *BRCA2*. Parental mouse ES cells (PL2F7) contain one conditional *Brca2* allele floxed by a *HPRT* minigene cassette and one deleted *Brca2* allele [35]. Southern blotting, Western blotting, qPCR and karyotyping was performed to validate the generation of the humanized-mouse ES cell line.

Figure prepared using a paid subscription to BioRender.com. **(b)** Southern blot showing the presence of *BRCA2* transgene (*Tg*) integrated into mESCs with conditional *Brca2 (cko = conditional knockout allele, ko = knockout Brca2 allele)*. F7 and F11 are two independent ES clones and F7 clone showed comparable intensity for *BRCA2* Tg, hence used for BRCA2 variant analysis. Probes used to detect the mouse *Brca2* conditional and knockout alleles and the human *BRCA2* transgene were designed as previously described [35]. **(c)** qPCR showing the expression of the relative copy number of *BRCA2* as compared to the single copy thymidine kinase (*TK* gene). qPCR probes are generated from 5', 3' and mid region of BRCA2. **(d)** Western blot showing the expression of BRCA2 in *Brca2^{cko/ko}* parental mESCs, *Brca2^{cko/ko}; Tg(BRCA2)* and in *Brca2^{ko/ko}; Tg(BRCA2)*. Vinculin is used as a loading control. **(e)** Chromosomal FISH showing the integration of *BRCA2* transgene on chromosome 12 (marked with spectrum orange probe), position A3 (marked with an arrow) **(f)** Immunofluorescence showing RAD51 (green) foci in response to 10Gy IR-induced DSBs in *Brca2^{cko/ko}* and *Brca2^{cko/ko}; Tg (BRCA2)* mESCs. γH2AX (magenta) is used as a marker for DSBs and nuclei were stained with DAPI (blue). n>50 cells quantified and represented as mean +/- SD in the associated graph. **(g)** EdU pulse labelling showing the doubling time between parental *Brca2^{cko/ko}* and *Brca2^{cko/ko}; Tg (BRCA2)* mESCs. n>50 cells quantified and represented as mean +/- SD in the associated graph. Figure prepared using a paid subscription to BioRender.com.

The frequency of each variant at day 3 post nucleofection was determined by amplicon sequencing of the genome-edited region using NGS. We found a strong correlation (Spearman's $\rho = 0.77$, $p < 1.14\ e^{-118}$) between the two independent replicates across exon 3, 18 and 19, suggesting good technical replicability of the approach (**S2B and S2C Fig**). The cutting efficiency of sgRNAs and the position of the mutations from the cut site also affects the variant generation and thereby read counts at day 3. One of the predominant pathways during CRISPR-Cas9 editing is the non-homologous end-joining (NHEJ) pathway that leads to the generation of indels (insertion and deletions). We found that indels leading to the loss of BRCA2 function were present at day 3, but eventually were lost from the pool by day 14 (**Figs 2B and S2D**). While we do not precisely understand how variants with a single nucleotide deletion in exon 3 are able to persist, it is worth noting that the majority of out-of-frame indels exhibit a significant fitness impact. The variants were also generated at variable frequency and the percentage of HDR ranges between 0.3% to 1.5% (**S2E Fig**). This suggests that loss-of-function variants leading to pathogenicity can be effectively classified using this approach.

Degenerate NNN oligos generates three nonsense variants TAA, TAG and TGA, as well as synonymous variants with no change in amino acid providing an internal control to investigate the effect of synonymous and non-sense variants on cell fitness. We next investigated the function scores (FSs) of these nonsense and synonymous variants. The FSs were calculated by taking $\log_2$ of the ratio of the frequency of NGS reads for a particular variant at day 14 to its initial frequency at day 3. The FSs of all the variants were bimodally distributed (F test for variance was calculated for each distribution, DMSO = 0.1062, Cisplatin = 0.1528, Olaparib = 0.1937, **Fig 2C**), and the nonsense variants had a more negative FS, whereas synonymous variants ranged between -1 to +0.6. This also validates the rationale to keep day 3 as the initial timepoint and day 14 as the end-timepoint of our assay, where we can accurately classify *BRCA2* variants. Interestingly, we found that changing residues at p.Asp2723 in exon 18 and p.Trp2788 in exon 19 led to more non-functional variants, whereas variants corresponding to residues at p.Tyr57, p.Glu58, p.Pro59 in exon 3 are overall functional (**Fig 2D**). This suggests the functional relevance of the key domains in BRCA2.

## A statistical classifier accurately classifies *BRCA2* variants based on cell fitness and their response to cisplatin and olaparib

*BRCA2* variants with a loss of function are expected to be depleted from the pool of cells by day 14 and can be regarded as pathogenic, whereas variants with a partial loss of function may not be significantly depleted by day 14. However, these latter variants are expected to exhibit sensitivity to DNA-damaging drugs, such as cisplatin and the PARP inhibitor olaparib. Functional variants (with no effect on BRCA2 function) would survive in the presence of the drugs, similar to the wildtype cells. We used FS of the variants for statistical modeling to calculate the probability of impact on function (PIF) to define pathogenicity.

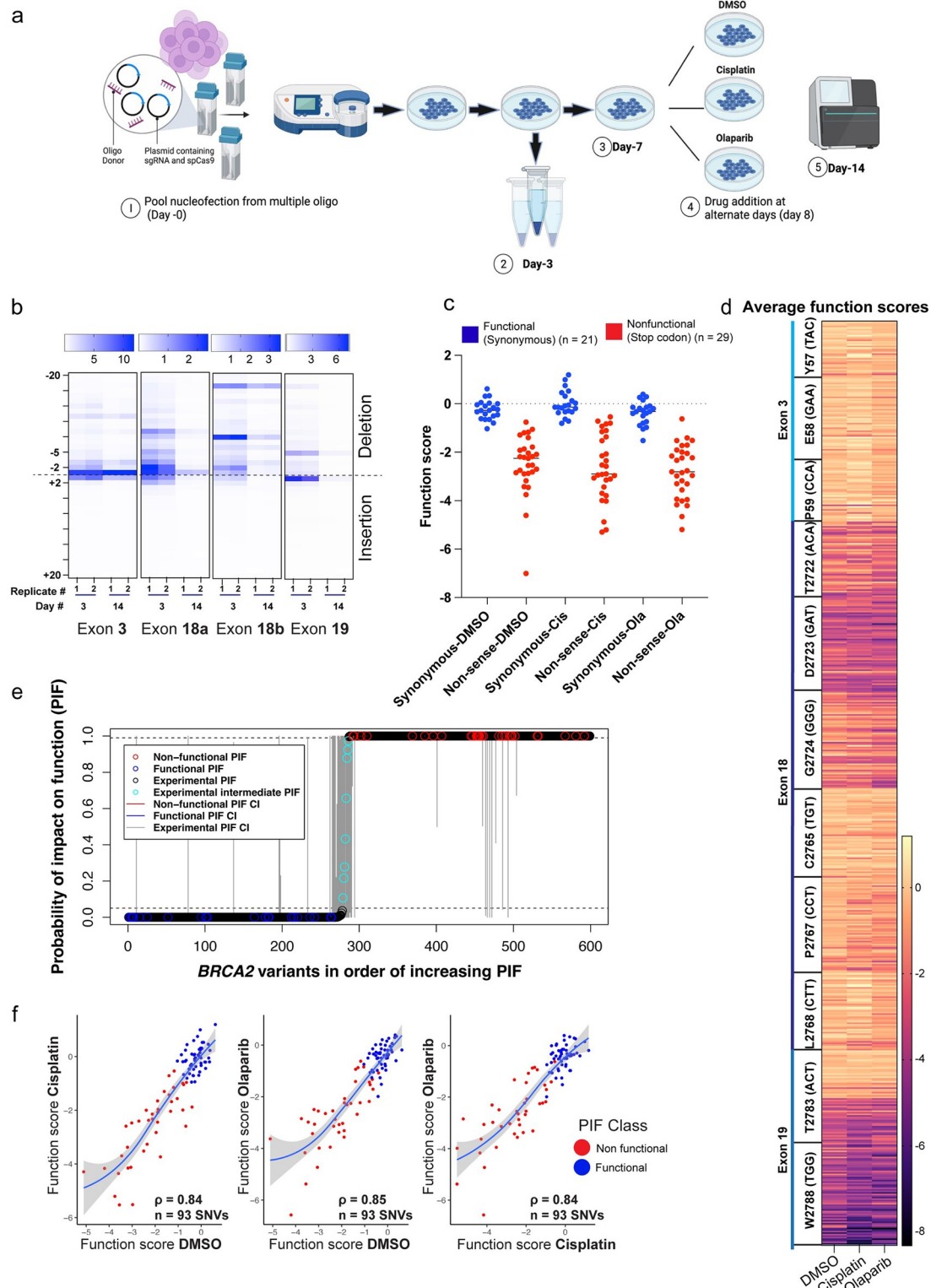

**Fig 2. Variant multiplexing using saturation genome editing (SGE) to inform classification of BRCA2 variants. (a)** Schematic overview of the experimental strategy to introduce *BRCA2* variants using a degenerate ssODN along with sgRNAs targeting a particular exon. The cells were collected at day 3 to confirm the generation of variants and at day 14 after selection with cisplatin and olaparib. Variant abundance was calculated using Next Generation Sequencing (NGS) and function scores were calculated. Figure prepared using a paid subscription to BioRender.com. **(b)** Heatmap showing the distribution of indels

formed at each exon. The frequency is normalized to the total read counts and z-score for each base pair indels is plotted. Each lane corresponds to the two independent replicates. The color-coded scale corresponds to the frequency of indels per exon. **(c)** Swarm plot showing normal distribution of function scores of 50 BRCA2 variants containing 29 nonsense and 21 synonymous variants (p value >0.05, Shapiro-Wilk Test) **(d)** The heatmap showing the distribution of function scores for 599 variants averaged from two independent replicates. The color scale ranges from light yellow (functional) to dark blue (non-functional) to depict enrichment and dropout scores of all variants in the presence of DMSO (control) and in the presence of cisplatin and olaparib. **(e)** Probability of impact on function (PIF) of 599 *BRCA2* variants from DMSO, cisplatin, and olaparib assays. The circles represent the PIFs for individual *BRCA2* variants, and the vertical lines show the 95% confidence intervals (CIs) for each PIF. The blue circles represent the synonymous variants and the red circles represent the nonsense variants. All other variants present in the oligo donor library are considered to be experimental. The dashed lines correspond to the functional-categorization thresholds of 0.05 and 0.99. This categorizes the experimental variants with PIF > 0.99 as non-functional, the variants with PIF ≤ 0.05 as functional, and the variants with 0.05 < PIF ≤ 0.99 as intermediate. **(f)** Strong correlation ($\rho$ = 0.84) between the function scores of *BRCA2* SNVs (n = 93) between DMSO, cisplatin and olaparib. Blue dots represent the functional and red dots are non-functional variants that were categorized according to our PIF-based calculation.

Statistical classifiers have previously been developed to estimate PIF for *BRCA2* variants [11,22,45]. Our earlier work utilizing a Bayesian hierarchical model (referred to as VarCall) has demonstrated the utility of integrating multiple functional assays to calculate the functional outcomes of different *BRCA2* variants [46]. Here, we use a novel statistical model that utilizes the FS values based on the cell survival (DMSO treated), cisplatin, and olaparib data to calculate PIFs for every variant in our full data set (N = 599), as described in Materials and Methods. The accuracy of the model on the full data set was 100%, whereas the K-fold cross-validation accuracy was lower but still sufficiently high (94%, 92%, and 92% for K = 5, 10, and 50, respectively). The cross-validation specificity and sensitivity varied depending on the value of K, but overall were close to the corresponding accuracy values. Indeed, for K = 5, 10, and 50, the <sensitivity, specificity> pairs were <92.5%, 93.3%>, <91.7%, 87.5%>, and <93.1%, 90.5%>, respectively. For each of the statistical models, the resultant PIFs were used to generate phenotype assignments for all the variants in the data set: PIF ≤ 0.05, PIF > 0.99 corresponded to functional, non-functional BRCA2 variants, respectively. The majority of the model-generated PIFs were robust, as reflected by their very narrow confidence intervals; most of the less-robust PIFs were in (or near) the central area of the plot, where the variants were in the intermediate zone (**Fig 2E**).

To assess the importance of including the FSs from all three assays (DMSO, cisplatin, and olaparib) in the model, we compared its performance with that of three one-variable models, each including only the FS from one of the assays (see Materials and Methods). While the one-variable models produced robust PIFs (**S3A–S3C Fig**), their accuracy on the full data set was notably less than 100%. Indeed, for DMSO-, cisplatin-, and olaparib-based models, the corresponding accuracies were 80%, 80%, and 70%, respectively. Moreover, the one-variable models (**S3A–S3C Fig**) resulted in a considerably larger number of variants in the intermediate zone and, therefore, much higher prediction ambiguity than the full, three-variable model (**Fig 2E**). We therefore concluded that the inclusion of all the three variables in the statistical model provides a substantial advantage in terms of model performance. The PIF score of variants falling into the "intermediate" zone represents the variants that our assay does not definitively classify as either functional or non-functional. Whether these variants inherently imply a hypomorphic nature of the variants needs further investigation in the future. We observed a strong FS correlation between DMSO and cisplatin, between DMSO and olaparib, and between cisplatin and olaparib, for all the *BRCA2* SNVs (Spearman's $\rho$ ranging from 0.84 to 0.85, n = 93 SNVs) (**Figs 2F and S4A and S4C**). Thus, out of the 599 *BRCA2* variants in our full data set, 313 variants were classified as non-functional and 278 variants were classified as functional. (**S4D Fig**). Only 8 variants fell into the intermediate zone where our model was unable to classify them into either classes, however, out of the 8 intermediate variants, two variants [p.

Pro2767Arg (c.8300_8301CT>GA) and pLeu2768Ser (c.8302_8303CT>AG)] have a PIF value of 0.97 and 0.92 respectively, which suggests that they are likely to be non-functional. The variants p.Glu58Val (c.173_174AA>TG), p.Gly2724Ser (c.8170_8172GGG>AGC), and p. Leu2768Pro (c.8303_8304TT>CC) each have a PIF value less than 0.27, suggesting that they are likely functional. In summary, plotting the average FS of 599 variants as a heatmap revealed a high probability of being functional variants at residue 57, 58, 59 in exon 3; 2765, 2767, 2768 in exon 18; whereas a high probability of being non-functional variants at residue 2722, 2723, 2724 in exon 18; and at residue 2788 in exon 19 (**Fig 2D**). The results obtained from functional assays are not always binary due to the cell-line-specific differences, technical or experimental variability. Hence, the development of a statistical classifier to generate a function-impairment probability, such as PIF is useful for the integration of results from multiple functional assays, providing a more reliable assessment of the impact of VUS.

## PIF-based variant categorization shows concordance with other computational metrics

Several computational metrics have been developed to define the pathogenicity of single-nucleotide variants. The potential clinical impact of *BRCA2* variants were analyzed using three computational predictors: PRIOR (http://priors.hci.utah.edu/PRIORS/), Combined Annotation Dependent Depletion (CADD) [47], and Bayes-del [48].

We found that 21 out of 31 SNVs (~68%) belonging to moderate (C35-C55) or high (C65) PRIOR class are also classified as non-functional in our PIF-based categorization. The remaining 10 variants also had a moderate to high PRIOR class but did not have any impact on function, of which three variants: p.C2765Y(c.8294G>A), p.P2767S (c.8299C>T), p.L2768P (c.8303T>C) were also previously reported to be neutral based on an HR assay[23]. Twelve out of 35 SNVs that belong to low PRIOR class (C0, C15, C25) were classified as non-functional according to our categorization. Four out of the 12 variants [p.D2723N (c.8167G>A), p. G2724V (c.8171G>T), p.W2788R (c.8362T>C), and p.W2788R (c.8362T>A)] were also previously reported to be non-functional [22,23]. Our PIF-based functional categorization using our three-variable statistical model shows high concordance with the PRIOR scores, with non-functional variants correlating with the "Extreme" PRIOR class and functional variants falling in the "Null" Prior class. (**S5A Fig**). CADD scores, which define the deleteriousness of SNVs in the human genome [47], show an inverse correlation with our FS. Pathogenic variants have a low CADD score in DMSO, cisplatin and olaparib samples (Spearman's $\rho$ = -0.59 in DMSO and cisplatin, $\rho$ = -0.57 in olaparib; N = 70 SNVs). Similarly, Bayes-del scores were also inversely correlated with our FS (Spearman's $\rho$ = -0.57 in DMSO, $\rho$ = -0.64 in cisplatin, $\rho$ = -0.63 in olaparib; N = 54 SNVs) (**S5B and S5C Fig**). While computational predictors may over-predict the probability of impact on function and similar discrepancies have been previously reported [22,46], we have used them in this study to support our categorization. We observed a good correlation between the function scores for variant's sensitivity to cisplatin and olaparib, we also observed 33 variants that have PIF scores <0.5 in DMSO samples but have a high PIF scores in cisplatin and olaparib. These variants survive with moderate growth disadvantage but are highly sensitive to DNA damaging drugs, suggesting that the incorporation of drug sensitivity arms allows enhanced functional categorization (**S5D Fig**).

Furthermore, our variant categorization is consistent with other functional assays performed by different laboratories using different cell lines, which gives us confidence in our functional categorization and can be used for clinical genetic testing (S2 and S3 Files). Notably, p.Thr2722Arg (c.8165C>G) and p.Asp2723His (c.8167G>C) in exon 18 are known non-functional controls that have been studied by several laboratories [21,23,49–51]. We have

categorized a ClinVar-reported VUS, p.Thr2722Ile (c.8165C>T), to be non-functional, which is consistent with a previously reported "likely pathogenic" categorization based on a multifactorial likelihood model [52]. Variants that change Asp at 2723 to Asn, Tyr, Ala, Gly, and Val are considered to be pathogenic [22,23]. The p.Asp2723Gly(c.8168A>G) affects splicing leading to exon 18 skipping, making this variant deleterious [53,54]. Our PIF-based functional categorization also revealed the deleteriousness of variants with Asn, Tyr, Ala, Gly, and Val at residue 2723. Moreover, we provide evidence that changing Asp at 2723 to Val and Glu also leads to pathogenicity (**Fig 3A**). *BRCA2* variants p.Cys2765Tyr (c.8294G>A), p.Pro2767Ser (c.8299C>T), and p.Leu2768Pro (c.8303T>C) that impact residues located in the oligosaccharide binding (OB1) domain, do not exhibit sensitivity to cisplatin and olaparib. These variants have been previously reported to be HR proficient [23]. Variant p.Pro2767Ser (c.8299C>T) was identified in a 26 year old young woman with breast fibrocystic dysplasia in a benign condition with no known pathogenic mutation in BRCA1 or BRCA2 genes [55]. While, a recent report suggested p.Pro2767Ser(c.8299C>T) to be pathogenic because it affects nuclear localization of BRCA2 and RAD51 foci formation [55], we found this variant to be functional. Our result is consistent with the previous report suggesting this variant to be functional based on its ability to perform HR [56] and no effect on splicing for this particular variant is reported from SpliceAI. Another variant, p.Leu2768His(c.8303T>A) has a high PRIOR probability (= 0.81) and is considered to be non-functional according to our functional categorization, but was reported to be neutral based on a HR-based functional assay [23] (**Fig 3A**). p.Trp2788Ser (c.8363G>C) in exon 19 has been previously shown to be pathogenic based on a HDR assay [23], and showed a likely pathogenic PIF In our previously reported BAC-based method [46]. We further validate that these variants are non-functional in this study. We also show that changing any nucleotide at residue 2788 to all possible non-wildtype nucleotides leads to a deleterious phenotype. P.Trp2788Arg(c.8362T>C) has been shown to be pathogenic in an HDR assay and is reported to be likely pathogenic in ClinVar. Currently, no information is available on the variants at residues 57, 58, and 59 that are identified in patients or reported in ClinVar. Our functional assay revealed that changing p.Tyr57, p.Glu58 and p.Pro59 to all possible combinations of SNVs does not affect BRCA2 function, and these variants are considered to be functional. The nonsense variants at positions p.Tyr57* (c.171C>A) and p.E58* (c.172G>T), identified in patients [57,58], were definitely non-functional based on our categorization.

## Targeted variant analysis validates SGE-based functional categorization

NGS-based analysis can suffer from a relatively low signal-to-noise ratio, leading to the possibility of sequencing errors. Therefore, independent validation is required to ascertain the reliability of these high-throughput assay. Moreover, CRISPR-Cas9-based SGE may introduce off-target mutations, which can affect other essential genes and thereby lead to cell lethality. Hence, we took a candidate-driven approach to generate some variants identified using our high-throughput SGE. We verified the effect of the missense BRCA2 variants on drug sensitivity and their ability to load RAD51 on DSBs during DNA repair. Since *BRCA2* variants that are non-functional will not survive in *Brca2*$^{ko/ko}$ mESCs, we have focused on the variants that are functional based on our PIF categorization. One of the advantages of using degenerate NNN oligos for targeted validation is the generation of multiple variants by a single round of electroporation. Here, we chose residue 2768 of exon 18 and utilized the NNN ssODN for this region to generate multiple variants for targeted validation. Several of the SNVs were validated based on computational predictions and published orthogonal functional assays (**Fig 3A**). Hence, we chose multi nucleotide variants, such as p.Leu2768Ser (c.8302_8303CT>TC), p.Leu2768Arg (c.8302_8304CTT>AGA) and *Brca2*$^{ko/ko}$-Tg(*BRCA2*) (wildtype) mESCs, along

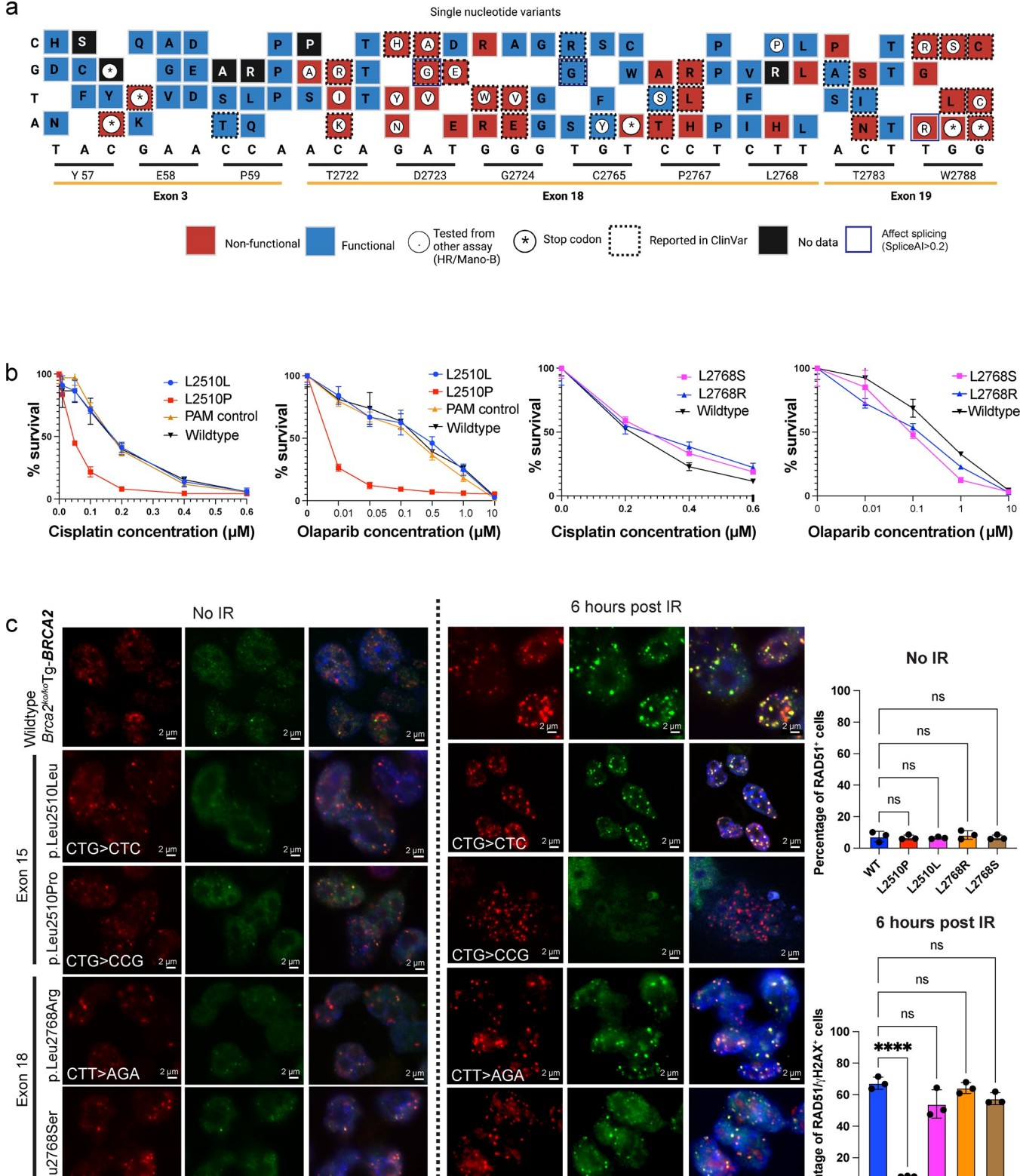

**Fig 3. Function scores derived from SGE accurately measures BRCA2 function. (a)** Sequence function map summarizing the variant categorization when each nucleotide is changed to other three non-wildtype sequences. Blue box represents functional and red box represents non-functional variant, star "*" represents nonsense variant, black box represents variants excluded due to low read counts, "C" represents confirmed based on functional assays using other cell lines (HR assay and MANO-B categorization [21–23,52]), black outline represents ClinVar reported variants and blue outline represent variants predicted to affect splicing based on SpliceAI score >0.2. The corresponding single letter amino acid change were denoted within in the box. Figure prepared using a paid subscription to BioRender.com. **(b)** Representative graphs showing the response of $Brca2^{ko/ko}$ mouse ES cells expressing *BRCA2* variants (p.Leu2510Leu, p. Leu2510Pro,p.Leu2768Arg,p.Leu2768Ser) to DNA damaging agents (cisplatin and olaparib. Values are based on XTT assay after 72 hours of treatment. PL2F7/F7 is a control expressing wildtype *BRCA2* in $Brca2^{ko/ko}$ mESCs and PAM control expresses only a synonymous mutation at the PAM site in exon 15 of BRCA2. **(c)** Representative image of *BRCA2* variants (p.Leu2510Leu, p.Leu2510Pro, p.Leu2768Arg, p.Leu2768Ser) showing RAD51 foci formation in unirradiated controls and after 6 hours post 10 Gy of IR. Double stranded breaks (DSBs) were marked by γ-H2AX in red and IR-induced RAD51 foci in green. Nucleus is stained with DAPI (blue), Scale bar = 5μm. **(d)** Quantification of the percentage of γ-H2AX$^+$ cells showing RAD51 foci. Results were expressed as mean+/ SD from three independent replicates where each dot represents a replicate with >100 cells quantified per replicate.

with controls such as only PAM modification for further validation. In addition, we have generated a previously characterized hypomorphic SNV p.Leu2510Pro (c.7529T>C) [36,59], along with a synonymous change p.Leu2510Leu (c.7530G>C) at the same residue 2510. Each of these variants also contains a synonymous change at the PAM site to block recutting by Cas9 after editing. We used a control where only the synonymous PAM modification was generated to examine whether PAM modifications alone would have any effects on BRCA2 function. We found p.Leu2510Pro(c.7529T>C) to be viable in $Brca2^{ko/ko}$ mESCs but highly sensitive to cisplatin and olaparib, which is consistent with our previous reports [60,61]. A synonymous change at position 2510, p.Leu2510Leu (c.7530G>C) and the variant with a PAM-only modification behaved like wildtype *BRCA2* (**Fig 3B**). p.Leu2510Pro also showed reduction in IR-induced RAD51 foci in comparison with p.Leu2510Leu and wildtype controls, suggesting impaired BRCA2 function due to Leucine to Proline change at residue 2510 (**Fig 3C**). Furthermore, we show that p.Leu2768Ser (c.8302_8303CT>TC) and p.Leu2768Arg (c.8302_8304CTT>AGA) does not impact its sensitivity to DNA-damaging drugs and maintains normal formation of IR-induced RAD51 foci (**Fig 3B and 3C**). This further validates that high-throughput CRISPR-based assays can be used to calculate PIF which accurately measures BRCA2 function. We anticipate that the combination of SGE coupled with drug sensitivity assays can be applied to other clinically actionable genes, thereby improving the clinical applicability of genetic data and contribute to the growing resource of MAVEs.

## A scalable approach to generate all possible missense variants of exon 13

Exon 13, encodes a critical region outside the C-terminal DNA binding domain of BRCA2 where several variants have been studied in functional assays and using biochemical studies [52,62]. Hence, we selected to functionally classify all possible SNVs from exon 13 for its structural significance, critical protein-protein interactions and functional domains. Specifically, we generated a ssODN oligo library containing all possible single nucleotide changes across exon 13 along with the synonymous PAM modification. sgRNAs were selected such that the PAM modification does not lead to change in amino acid and do not affect splicing. The entire exon 13 was divided into two parts from c.6938-12A in the intron and from residue p.Gly2313 (c.6938G) to p.His2324(c.6972T) using sgRNA 13.2 and 13.3 (experiment# 13.23) and from p. Val2325(c.6973G) to p.Arg2336(c.7007G) to c.7007+7 position into the intron using sgRNA13.2 (experiment #13.2) (**Fig 4A**). We observed a good correlation in read counts suggesting SNVs were generated at a similar frequency, and only 12 SNVs out of 264 possible missense variants were excluded due to low read counts and discordance between the replicates (**S6A Fig**). The percentage of HDR in this region varies between 0.6 to 2.3% and the out of frame indels were eventually lost at day 14 suggesting loss of function variants in this region is deleterious (**S6B and S6C Fig**). We also observed a strong correlation between the function

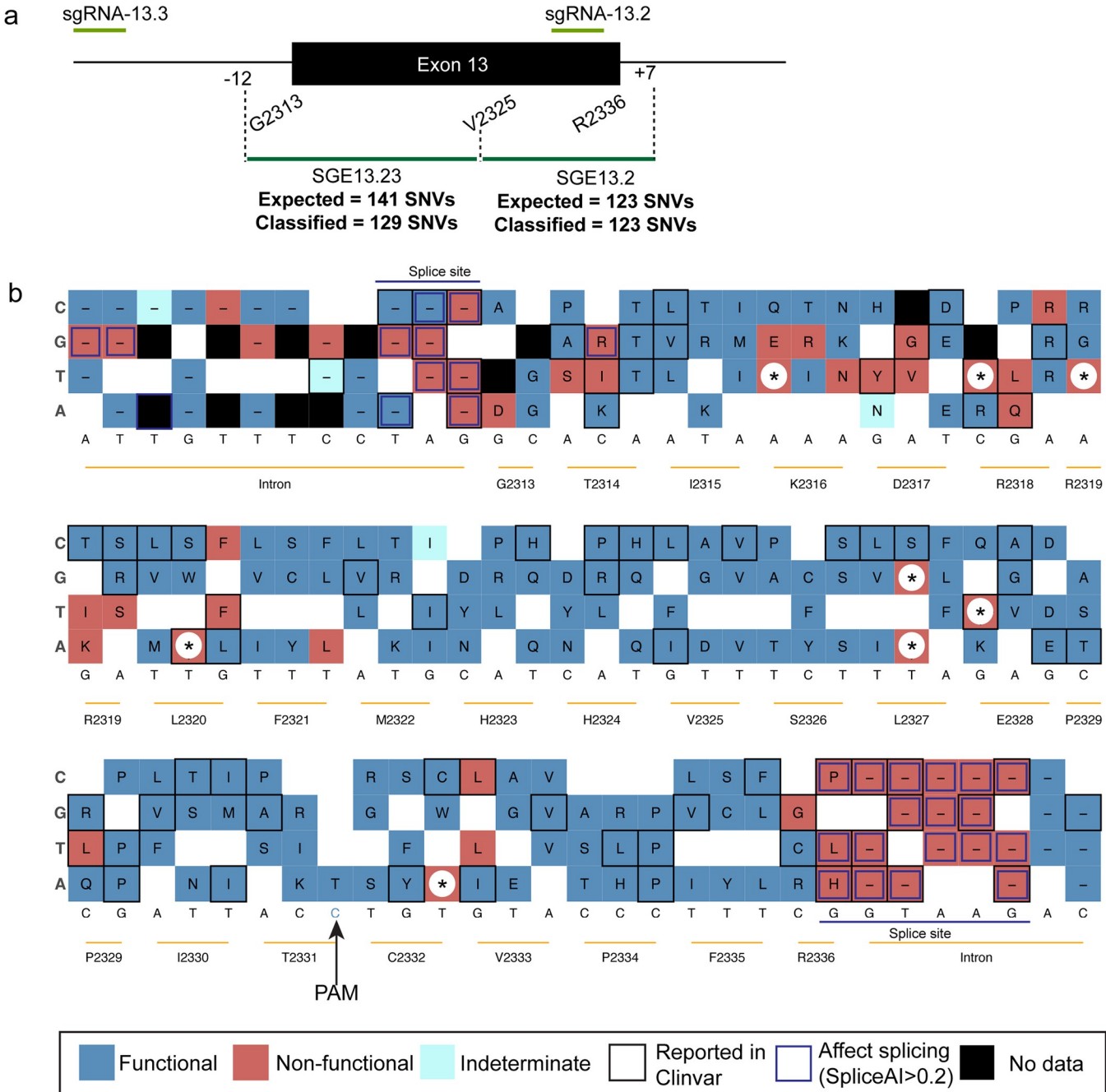

**Fig 4. Saturation genome editing of all possible single-nucleotide changes of exon 13. (a)** Experimental strategy showing two sgRNAs were used to saturate all possible SNVs in exon 13. sgRNA13.2 and sgRNA13.3 were used to saturate from -12 position and residues 2313 to 2325 and 129 SNVs were categorized out of 141 possible SNVs. sgRNA13.2 was used to saturate residues 2325 to 2336 and +7 into the intron and 123 out of 123 possible SNVs were categorized. **(b)** Sequence function map summarizing the variant categorization when each nucleotide is changed to other three non-wildtype sequences across entire exon 13. Blue box represents functional and red box represents non-functional variant, star "*" represents nonsense variant, black box represents variants excluded due to low read counts and discordance between the replicates, black outline represents ClinVar reported variants and blue outline represent variants predicted to affect splicing based on SpliceAI score >0.2. The corresponding single letter amino acid change were denoted within in the box.

scores in all the three conditions (DMSO, cisplatin and olaparib) with spearman rho ranging between 0.81 to 0.89 (**S6D Fig**). We observed a bimodal distribution of the function scores for all the SNVs with 42 out of 43 synonymous variants to be functional and 8 nonsense variants

behaving as non-functional (**S6E Fig**). Using the same PIF threshold and Gaussian mixture-modelling we can categorize 188 SNVs to be functional (74.6%), 60 SNVs (23.8%) to be non-functional and only 4 variants were categorized as indeterminate (1.5%) (**S6F and S6G Fig**). The function scores for functional category ranges between 0.57 to 1.75 and the non-functional category of variants ranges from 0.53 to -1.15 (S2 **File**). The mixture-models revealed 100% sensitivity and 95.45% specificity of variant classification, and the positive likelihood ratio (LR+) of our dataset is 22 and the negative likelihood ratio (LR-) is 0. Furthermore, to estimate the evidence strength of our assay, we used the 43 synonymous and 8 non-sense SNVs from exon 13 as the training dataset and our model exhibited 85.7% sensitivity and 100% specificity in classifying ClinVar reported variants (**S6H Fig**). Based on the recommendations for application of the functional evidence PS3/BS3 criterion using ACMG variant interpretation framework [63,64], we calculated the Odds of Pathogenicity (OddsPath) value of our model. Our OddsPath value for pathogenic variant is 9.52 supporting PS3_moderate level of evidence and 0.06 for benign variants supporting BS3_moderate level of evidence. We also observed a negative correlation between the function score of the SNVs in our assay and some of the other computational predictors such as BayesDel (R = -0.23, p = 0.00019) and CADD (R = -0.32, p = 3.1 e-7) suggesting strong overlap between our functional categorization and these prediction models (**S6I Fig**). Furthermore, summarizing our entire data set as a sequence-function map revealed the splice site variants at c.7007+1G>T, c.7007+1G>C, c.7007+1G>A, p.Arg2336(c.7007G>A, c.7007G>C, c.7007G>T) were mostly non-functional (**Fig 4B**). The variant p.Arg2336Pro (c.7007G>C) and p.Arg2336Gly (c.7006C>G) were classified as neutral in a cDNA-based functional assay [21]. However, our previous studies reported p.Arg2336Pro (c.7007G>C) to be non-functional and affect splicing [46], which is in agreement with its pathogenic classification in ClinVar. This further supports the advantage of our assay that can consider the potential effect of the variants on splicing, as opposed to other cDNA-based functional assays. p.Arg2336Leu(c.7007G>T), another variant that affects splicing and is categorized as non-functional and is "likely pathogenic" in ClinVar. We have not observed any discordances with the variants that scored pathogenic/likely pathogenic and benign/likely benign classification in ClinVar except a synonymous variant p.Arg2318Arg (c.6954A>C) and p.Arg2318Gln (c.6953G>A) which is considered to be benign [52] but is categorized as non-functional in our assay. However, a recent report further characterized this variant using biochemical manipulations and showed the inability of this variant to interact with HSF2B critical for BRCA2 function [62]. Future studies focused on mechanistic validation of these two variants will reveal the impact of these SNVs, which we speculate to be affecting splicing. Similarly, p.Pro2329Leu(c.6986C>T) also showed reduced interaction with HSF2B and is categorized as non-functional, but p.Pro2334Leu(c.7001C>T), which exhibited normal interaction is functional in our assay. Across the entire exon13, we observed a large fraction of variants to be functional except region from residues 2316 to 2320 where majority of the non-functional variants were scored. In this region, p.Arg2319Thr(c.6956G>C) is considered to be a VUS with a probability of 0.05 [52] is categorized as a functional variant in our assay. Similarly, p.Thr2331Ala(c.6991 A>G) and p.Cys2332Tyr(c.6995 G>A) were also considered to be likely benign [52] and is scored as functional variant in our dataset. Based on our findings, we propose that a region spanning 50–100 nucleotides can effectively be saturated using a pair of sgRNAs. This suggests that any region of BRCA2 could potentially be saturated, contingent upon the availability of sgRNAs at intervals of 30–50 nucleotides. For instance, the C-terminal DNA binding domain of BRCA2, encoded by exon 15 to 26 and spanning 2213 nucleotides, hosts many pathogenic missense variants. We predict that this region could be comprehensively saturated for all possible SNVs using approximately 50 oligo donor libraries in conjunction with a single or a pair of sgRNAs. In conclusion, the SGE data of exon 13

establishes the scalability of our cloning-free oligo donor library-based SGE and lays the foundation for saturation of the entire *BRCA2* coding sequence.

## Materials and methods

### 1. Maintenance of mouse embryonic stem cells

Mouse embryonic stem cells (parental clone of PL2F7 cells [35]) were cultured with underlying feeder cells (SNLP) that express leukemia inhibitory factor (Lif) and puromycin resistance cassette. The maintenance medium consisted of Knockout DMEM (Gibco) supplemented with 15% FBS (HyClone), penicillin-streptomycin-glutamine (Gibco) and 0.1 mM β-mercaptoethanol. The media was changed every day and cells were trypsinized when ESCs reached 80% confluency.

### 2. Generation of mouse embryonic stem cells with a single copy of human BRCA2

To generate the humanized mouse ES cells, we electroporated 20 μg of linearized BAC DNA (clone # CTD-2342K5 with a 127 kb insert containing full-length *BRCA2)* into $1.0 \times 10^7$ PL2F7 mESCs and selected in the presence of G418 (Invitrogen) as previously described [35]. Neomycin (G418) was used to select mESC for the presence of BAC containing full length wildtype *BRCA2*. After the BAC-containing mESC were obtained, the presence of the wildtype *BRCA2* was confirmed by Sanger sequencing. PGK-CRE plasmid was electroporated to $10^7$ PL2F7 mESCs containing a single copy of *BRCA2*. A transient Cre expression resulted in loss of the *cko* allele and generation of mESCs that lack endogenous *Brca2* but make a functional *HPRT* minigene. The cells were selected in the presence of HAT media followed by HT media to enrich for cells with functional HPRT and functional complementation of *Brca2* lethality by the expression of BRCA2. We also confirmed the presence of the *BRCA2* transgene by Southern analysis using probes described previously [35], expression at the protein level by Western blot and single copy integration was confirmed using qPCR. Rabbit polyclonal BRCA2 antibody (BETHYL lab, Cat # A303-434-A-T, 1:2000 dilution) was used to detect the expression of BRCA2 in the mESCs and mouse monoclonal Vinculin antibody (Santa Cruz biotech, Cat# sc25336, 1:200,000 dilution) as a loading control. ECL plus Western blotting detection system (Amersham) was used for chemiluminescent detection. Southern blot and Western blot were performed based on methods previously described [36]. The clonal population containing single copy of BRCA2 are designated as **PL2F7/F7** and were used for all the experiments.

### 3. Spectral karyotyping and chromosome FISH

PL2F7/F7 mESCs were arrested at metaphase by incubating the cells with Colcemid (15210–040, KaryoMAX Colcemid Solution, Invitrogen, Carlsbad, Calif., USA) (10ug/ml) for 3 hours prior to harvest. Cells were collected and treated with hypotonic solution (KCL 0.075M,6858–04, Macron Chemical) for 15 minutes at 37°C and fixed with methanol: acetic acid 3:1. Slides were prepared and kept overnight for use in SKY analysis. The bacterial artificial chromosome (BAC) probe corresponding to *BRCA2* was labeled with Nick Translation using Fluorescein-12-dUTP (green) (Enzo life sciences, ENZ-42910). In-situ hybridizations of the probes were performed with about 300–400 ng of each NT product per probe with 10 μg of mouse Cot-1 were precipitated and then dissolved in 10μl hybridization buffer (formamide 50%, dextran sulfate 10%, 2 x SSC). The probe and prepared slide were co-denatured at 75°C for 5 min and hybridized in a humidity chamber at 37°C overnight. Post-hybridization washes followed standard procedures at 45°C. Confirmation of chromosomal location was carried out using a

mouse whole chromosome paint for chromosome 12 in Spectrum orange (Metasystems, D-1412-050-OR). Counterstaining was performed with Vectashield mounting medium with DAPI (H-1200, Vectashield). *BRCA2* transgene was confirmed to be located at Chr12:A3.

## 4. Guide RNA design and cloning

20 bp of crRNA sequences were designed using Benchling and sgRNAscorer 2.0 [65] for high on-target score and based on the ability to generate a synonymous PAM modification. SpliceAI-based analysis was performed to ensure that the synonymous PAM modification didn't induce any splicing changes. The single guide RNAs (sgRNAs) were cloned into pX330 vector, that expresses the gRNA from a U6 promoter and Cas9 expression cassette. Complimentary oligonucleotides were ordered from Integrated DNA Technologies (IDT) and were annealed, phosphorylated, diluted, and ligated into *Bbs*I-digested pX330 as previously described [66]. sgRNAs were selected based on the availability to introduce synonymous PAM mutations. The plasmids were confirmed by Sanger sequencing using U6 primers for the correct integration of crRNA sequences. Single-stranded oligodeoxynucleotide (ssODN) were ordered from IDT as a 180 bases long ultramer. The oligo contains a fixed synonymous PAM mutation that prevents recutting by the Cas9 and acts as a fixed HDR marker for sequencing analysis. Each oligo contains a degenerate codon (NNN) for each position that will give rise to all the possible combination of nucleotides giving rise to $4^3$ variants (63 non-wildtype combinations).

## 5. Nucleofection for CRISPR-Cas9 based saturation genome editing

mESCs were trypsinized and plated in a 10 cm culture dish a day before nucleofection to promote cells to be in actively dividing phase. Three micrograms of plasmid (containing desired sgRNA and Cas9-expressing cassette) and 6 µg of ssODN oligo was nucleofected into $3\times10^7$ ESCs using a Lonza nucleofector 2B as per manufacturer's recommendation (Program: A030). We have used the clonal population of PL2F7/F7 mESC line for each independent replicate. Three separate nucleofections were performed using the ssODN and the nucleofected cells were then combined into one 10cm feeder plate, which we referred to as replicate-1. Similarly, PL2F7/F7 cells were grown and the same procedure was performed as another independent replicate (replicate-2) with the same pair of sgRNA and oligo. For each of the replicates, the cells were cultured for 72 hours with media changes every 24 hours. After 72 hours, each plate was trypsinized and half of the cell population was pelleted for DNA isolation with the remaining cell population re-plated onto one 10cm feeder plate and cultured for 4 more days with daily media changes. At day 7 post nucleofection, each plate was trypsinized and $10^7$ cells were plated onto three 10 cm feeder dishes for the drug treatment. On day 8,10 and 12, media with 0.4 µM cisplatin or 0.05 µM PARP inhibitor (olaparib), respectively, was added to the 10 cm dishes along with a DMSO control. Cells were pelleted down at day 14 from DMSO, cisplatin and olaparib treated dishes for DNA extraction.

## 6. Genomic DNA isolation for Next generation Sequencing (NGS)

Genomic DNA was extracted from the cell pellets using Zymo Genomic DNA Extraction kit (Cat# D3024). Twenty micrograms of DNA was used for PCR amplification using exon specific primer and Platinum Taq High Fidelity Polymerase (Invitrogen, Cat# 11304029) to amplify the region of interest. Ninety-six PCR reactions for each sample were performed and the PCR products were pooled together and then purified using QIAquick PCR purification kit (Cat # 28106) The size of the PCR product was confirmed using agarose gel electrophoresis and samples were sent to Azenta/Genewiz (New Jersey) for NGS. DNA libraries were prepared

using TruSeq Nano DNA Library Prep kit (Illumina) according to the manufacturer's protocol and were deep sequenced using MiSeq (2x250 cycle).

## 7. Variant analysis and functional categorization

Paired-end sequencing was performed on an Illumina MiSeq instrument with a read coverage of 1–3 million reads per sample. The reads were demultiplexed using bcl2fastq (Illumina) and fastq files were generated. The paired end reads were merged using FLASH and reads were aligned to the reference amplicon with a global sequence alignment algorithm using Needleman-Wunsch Alignment. The total number of aligned reads were quantified using CRISPResso2 http://crispresso.pinellolab.org/submission [67] and individual unique alignments were annotated with a custom-made variant caller modified from ANNOVAR [68] (release 2019-10-21) and the R/Bioconductor Biostrings package in R version 4.1.2 https://bioconductor.org/packages/Biostrings. Merged reads containing "N" bases were removed from the analysis. Abundances of SNVs were quantified when the reads contained the synonymous PAM modification (HDR marker) and no other substitution or indels were present. The variants generated at a low frequency (less than 1 in $10^5$ reads in any one of the replicates) were excluded from further analysis to prevent erroneous variant categorization. A pseudo count of 1 was added to all reads and for every sample and at all the conditions. Read counts for each SNVs were then normalized to the total read coverage of the sequencing library. Individual variants with a read count of more than 1 in 100,000 reads at day 3 were used for analysis. Dropout or enrichment scores were calculated by taking the ratio of frequency of the variants at day 14 in DMSO or cisplatin or olaparib over day 3. The scores were expressed in $Log_2$ scale, which we define as function scores of SNVs in DMSO, cisplatin and olaparib. The FSs, averaged over two independent replicates, were fit using a probit regression model for statistical modelling to calculate the probabilities of impact on function (PIFs; see the next subsection).

## 8. Statistical modeling to calculate the PIFs

We regarded the computational assignment of phenotype (i.e., functional or non-functional) to the *BRCA2* variants in the full dataset (N = 599) as a binary categorization problem on the space of all such variants. We followed the paradigm of supervised learning and used the variants with a known phenotype (21 synonymous and 29 nonsense variants) were used for the training set. On this training set, we fit a linear, three-variable probit regression model whose three predictors were the FSs values for the DMSO, cisplatin, and olaparib assays. The resultant model was applied to the entire data set. For each *BRCA2* variant, the three-assay PIF was defined as the model-generated probability of being non-functional. Additionally, we analyzed three one-variable linear probit regression models, where the predictors were the same three variables considered independently. The one-variable models were trained the same way and yielded one-assay PIFs when applied to the entire data set.

The accuracy of a statistical model on the full data set was defined as the fraction of correct phenotype assignments on the training set (i.e., the model-fit accuracy). In addition to this accuracy, we assessed the predictive ability of each model via K-fold cross-validation with K = 5, 10, 50. In this approach, a data set is (in our case, the 50 *BRCA2* variants with a known phenotype) is randomly divided into K equal-size parts, or "folds". One of these folds is used as a test set, and the remaining K–1 folds, combined, are used as the training set. The statistical model is trained on the training set, and then its predictive accuracy is calculated on the test set. This process is repeated for each of the K folds. The values K = 5 and K = 10 for the fold numbers are commonly used values, and K = 50 corresponds to the frequently used leave-one-

out cross-validation [69]. For each fold, predictive accuracy was calculated as the fraction of correct phenotype assignments on that fold. The overall accuracy of K-fold cross-validation was calculated as the predictive accuracy on each fold, averaged across all the K folds. Besides the accuracy, we also computed sensitivity and specificity for each cross-validation case. For K-fold cross-validation, sensitivity and specificity were defined as the fractions of all non-functional variants and all functional variants in the test set, respectively, whose phenotypes were predicted correctly, averaged across the number of folds included in the analysis [69].

To assess the robustness of the PIFs generated for the full data set (N = 599), we calculated 95% confidence intervals for each PIF value. To this end, for each of our four probit-regression models, we ran 10,000 simulations. In each simulation, each data point for each predictor variable in the training set (N = 50) was additively perturbed by a random, independently generated, normally distributed number with zero mean and a standard deviation of 0.05 (see, e.g. the "small noise" randomization strategy in Ref [70]). The probit models were then refit to these perturbed data sets, and the PIFs for the full data set were recalculated. As a result, for each PIF, we obtained a sample of 10,000 values representing the PIF's random variation on the perturbed data. From each such sample, the confidence interval for the PIF was computed using the percentile method [71]. The PIFs and corresponding confidence intervals for each variant are provided in the S2 File. The variability in FS was simulated computationally, rather than based directly on repeats of the experimental assay for each variant (indeed, by design, the PIF-calculation algorithm worked on repeat-averaged data). Note that, for a few of the variants, the corresponding CIs were of considerable size. The occurrence of a few large CIs could be expected and, during our statistical-model development, was associated with our model's ability to perform sharp separation between functional and non-functional variants with the stringent PIF thresholds. For a few variants, this resulted in an increased sensitivity of the corresponding PIFs to the training-data randomization used in CI generation.

All the code pertaining to PIF calculation and analysis was written in R 4.0.2 (2020-06-22), using RStudio ver. 1.3.1073, and was developed and run on a Dell Latitude 7400 laptop computer with an Intel Core i7-8665U CPU @ 1.90GHz processor and 16.0 Gb RAM under the 64-bit Microsoft Windows 10 Enterprise ver. 20H2.

## 9. SGE of all possible SNVs from exon 13

SNVs with day 3 read counts below 10 or day 3 frequency below 1 read in 1e5 were filtered. A first analysis was carried out for each replicate independently and for each condition (DMSO, Olaparib, Cisplatin). SNVs with a post-pre ratio (day 14 over day 3 ratio) difference of more than 2 between both DMSO replicates and leading to divergent results for functional class assignment for these replicates were filtered. SNVs with a post-pre ratio difference of more than 2 between replicates in at least two conditions (for example between cisplatin replicates and between olaparib replicates) and showing divergent results for functional class assignment in these two conditions were also filtered. For each condition (DMSO, cisplatin, olaparib), the post-pre ratio is calculated using the mean of the post-pre ratio of the two replicates. Then, a global post-pre ratio is calculated using the weighted mean of the mean post-pre ratio of each condition (weights: 0.4 for DMSO, 0.25 for cisplatin and olaparib).

Position biases in editing rates were modelled using day 14 over day 3 ratios and chromosomal position for each SNV and within each exon. The "loess" function from R was used to fit the log2 of this ratio as a function of chromosomal position. Only the SNVs with a ratio 1.25 were used to limit influence of biological effect in the model. The model thus provided a score for each chromosomal position corresponding to position biases. For each SNV, this score was subtracted from the log2 of the day 14 over day 3 ratio.

The corrected log2 day 14 over day 3 ratios were linearly scaled within each exon. Then, the obtained function score was normalized across all exons such that the median synonymous and nonsense function scores within an exon were the same as the global scores across all exons. A Gaussian mixture model was used to estimate the probability for each SNV to be pathogenic or benign, using its function score. All non-sense variants and all variants classified as "Pathogenic" or "Likely pathogenic" in ClinVar database constituted the "Non-functional" class in the training dataset. All synonymous and all variants classified as "Benign" or "Likely benign" in ClinVar database constituted the "Functional" class. 42 known functional and 12 known non-functional variants were used for the training dataset. Using this training dataset, the "mclust" function in R generated and trained a model to predict pathogenicity of a SNV using its function score. The model then calculated the probability for each SNV to be pathogenic. SNVs with a probability higher than 0.99 were considered "Pathogenic" whereas those with a probability lower than 0.05 were considered "Benign". SNVs that had a probability falling between 0.05 and 0.99 were classified as "Intermediate" where our model was unable to classify them into either classes. We calculated positive Likelihood Ratio (LR+) and negative Likelihood Ratio (LR-) to provide an indication of the probability of a variant being pathogenic based on our functional assay results. The positive LR represents the likelihood of a variant being truly pathogenic, while the negative LR represents the likelihood of a variant being incorrectly assigned as benign, when it is actually pathogenic. It is calculated by the following formula:

Positive LR = sensitivity / (100 –specificity).

Negative LR = (100 –sensitivity) / specificity.

We used the synonymous and non-sense variants as cutoff for functional and non-functional variants and calculated OddsPath for pathogenic and benign variants using the following formula for evidence of strength:

P1 = Number of pathogenic control variants / Total number of control variants

P2_pathogenic = Number of pathogenic variants predicted / (Number of pathogenic variants predicted + 1)

P2_benign = 1 / (Number of benign variants predicted + 1)

OP_pathogenic = (P2_pathogenic * (1—P1)) / ((1—P2_pathogenic) * P1)

OP_benign = (P2_benign * (1—P1)) / ((1—P2_benign) * P1)

The OddsPath was then equated with corresponding level of evidence strength according to the Bayesian adaptation of the ACMG/AMP variant interpretation guidelines [63,64].

## 10. SpliceAI-based analysis

We used SpliceAI, a deep learning-based tool [41] to evaluate the impact of each SNV on the risk of splicing events due to the PAM modifications. The prediction was used to investigate if the PAM modification alone impacts splicing or not. The reference sequence with the synonymous PAM modification was then used as an input to assess the impact of each SNV in the context of the PAM-mutated sequence. For each possible SNV, the tool returned the probability of increasing and decreasing risk of splicing events at strategic positions. SpliceAI score > 0.2 is used to assess the potential impact of variants on splicing [42].

## 11. Immunofluorescence and confocal imaging

Approximately 5 x $10^4$ mouse ES cells were seeded on poly-D-Lysine coated coverslips (neuvitro GG-12) and cells were irradiated (IR) with 10Gy of γ-radiation after 24 hours. After 3 hours post IR, the cells were treated with hypotonic solution (85.5 mM NaCl, 5mM MgCl2, pH 7) for 10 min followed by fixing solution (4% Paraformaldehyde, 10% SDS in PBS) for

10min. Cells were incubated overnight at 4˚C with primary antibodies: γH2AX (1:500, Millipore JBW301), RAD51 (1:250, Millipore PC130) diluted in antibody dilution buffer (1% BSA, 0.3% TritonX100, 5% goat serum in PBS). The following day, cells were washed three times with PBS containing 0.2% Triton X 100 (PBST) and incubated with secondary antibodies (Alexa-fluor anti-mouse 594 [Invitrogen A11005], and anti-rabbit 488 [Invitrogen A11034]; 1:500) diluted in PBS at 37˚C for 1 h. The cells were washed three times with PBST and stained for DAPI (1:50,000, Sigma 11190301) for 5 min. The coverslips were mounted on clean labeled slides with anti-fade mount (Invitrogen P36930) and imaged in Zeiss 710 confocal microscope (63X oil immersion objective) and analyzed utilizing Carl Zeiss zenblue 2.6 software.

## 12. Generation of individual BRCA2 variants in PL2F7/F7 mESCs

For the generation of individual variants of BRCA2, we used the PL2F7/F7 mESC line that contains a single copy of *BRCA2*. Individual variants were generated by CRISPR-Cas9 based knock-in approach. Three micrograms of guide RNA plasmid (Px458; Cas9-2A-GFP) and 6μg of ssODN oligo was nucleofected into $3x10^7$ ESCs using a Lonza nucleofector 2B as per manufacturer's recommendation (Program: A030). After 48 hours, the cells were trypsinized and $2x10^4$ GFP+ cells were isolated using FACS. The cells were plated in a 10 cm dish containing SNLP feeder cells. After 7 days, colonies started to grow, and 96 colonies were isolated and individually plated in a 96 well plate containing SNLP feeder cells. The cells were duplicated into two 96 well plates when they were confluent. One plate was processed for genomic DNA isolation to identify the mutation and another plate for maintaining the clonal population. The region of interest was PCR amplified and confirmed for the presence of the desired mutation and absence of any additional mutations by Sanger sequencing.

## 13. Drug sensitivity assay

mESC clones containing individual variants were thawed from a replicate 96 well plate and expanded to 2x24 wells of a 24 well plate. For the drug assay, early passage clones were plated into wells of a gelatinized 96 well plate at 12,000 cells/well in 200μl of ESC maintenance media containing Knockout DMEM (with 15% FBS, 1x β-mercaptoethanol and 1x Penicillin/Streptomycin/Glutamine). After 24 hours, triplicate wells were replaced with 200 μl each with olaparib (0, 0.01, 0.05, 0.1, 0.5, 1.0, 10 μM) or cisplatin (0, 0.1, 0.2, 0.4, 0.6 μM). The plates were incubated for 72 hours, and XTT cell viability assay was performed as previously described [35].

## Supporting information

**S1 Fig. Experimental schematic showing the rationale to generate mouse ES cells expressing single copy of BRCA2 and generation of BRCA2 variants using saturation genome editing (SGE).** CRISPR-Cas9 based editing along with a pool of donor DNA containing NNN degenerate nucleotides, directly into the integrated *BRCA2* transgene (Tg) allows generation of *BRCA2* variants. The generation of indels is also limited as loss of function variants will be lost from the pool and only variants generated by HDR will survive. All HDR variants that are generated by CRISPR-Cas9 SGE will be present at the initial time point at day 3 and loss of function variants (pathogenic) will be eventually lost by day 14. Hypomorphic variants with partial loss of BRCA2 function are sensitive to DNA damaging drugs, cisplatin and PARP inhibitor (olaparib), thereby distinguishing between hypomorphic and neutral variants. Figure prepared using a paid subscription to BioRender.com.
(TIF)

**S2 Fig. BRCA2 variants can be reliably generated by CRISPR-based SGE and loss-of-function variants deplete from the pool over time. (a)** Schematic representation of 3418 amino acid long BRCA2 showing its key functional domain. Variants were generated at residue 57, 58, 59 of Exon 3 and 2722, 2723, 2724, 2765, 2767, 2768 of exon 18 and 2783,2788 of exon19. **(b)** Pearson correlation showing the read abundance for each variant at day 3 between two independent replicates. The data is represented in log10 scale, $\rho = 0.77$ for 599 BRCA2 variants. Each exon is color-coded. The read counts were strongly correlated between individual exons ($\rho = 0.79$ for 130 variants of Exon3, $\rho = 0.77$ for 343 variants of Exon 18 and $\rho = 0.90$ for 126 variants of Exon 19) **(c)** Histogram showing the distribution of read abundance for each replicate and the average of the read abundance. **(d)** Graph representing the distribution of indels from the cut site "0" for each exon. The values were expressed as percentage of reads normalized to the total number of reads for each exon at day 3 and at day 14. Blue line represents day 3 and red line represents day 14. **(e)** Quantification showing the percentage of indels and percentage of HDR between individual exons. Each dot represents an independent replicate.
(TIF)

**S3 Fig. Development of a statistical classifier to classify BRCA2 variants.** Probability of impact on function (PIF) for the *BRCA2* variants in the full data set (N = 599), calculated using **(a)** only DMSO-assay data **(b)** only cisplatin-assay data **(c)** only olaparib-assay data. The circles represent individual *BRCA2* variants, and the vertical lines show the 95% confidence intervals (CIs) for each PIF (for some of the variants, the confidence intervals are negligibly small). The dashed lines correspond to the functional-classification thresholds of 0.05 and 0.99. Most of the black circles are positioned so close that they form a continuous thick black line.
(TIF)

**S4 Fig. Development of a statistical classifier to calculate probability of impact on function (PIF). (a-c)** A strong correlation of the function scores between two independent replicates for 599 variants (r>0.7) for DMSO, cisplatin and olaparib samples. Red dots represent nonfunctional, blue dot represents functional and gray dot represents intermediate category of variants. **(d)** Histogram showing the distribution of 599 variants of which 29 variants are known to be non-functional as it encodes for a stop codon, 21 variants are known to be functional as it leads to a synonymous change in nucleotide with no change in amino acid. Remaining 549 variants were experimental and are bimodally distributed in DMSO, cisplatin and olaparib samples. The known functional and nonfunctional variants were used as a training dataset to develop the statistical classifier. **(e)** The probability of impact on function (PIF) was calculated for all the 549 experimental variants. The histogram shows the distribution of PIF-classified functional (278 variants), nonfunctional (313 variants) and 8 variants that falls into the intermediate zone were denoted by an asterisk.
(TIF)

**S5 Fig. PIF-based functional categorization of *BRCA2* SNVs accurately predicts in-silico predictors. (a)** Strong correlation of function scores between DMSO and cisplatin, DMSO and olaparib, cisplatin and olaparib samples showing the distribution of *BRCA2* SNVs labelled based on PRIOR pathogenicity class. **(b-c)** Negative correlation between SGE-derived function scores of single nucleotide variants (SNVs) across 11 codons in DMSO, cisplatin and olaparib with **(b)** CADD scores ($\rho = -0.57$ to $-0.60$, n = 71 SNVs) and **(d)** Bayes-del score ($\rho = -0.6$ to $-0.66$, n = 71 SNVs). Blue dots represent the functional and red dots are non-functional variants according to our PIF-based calculation. **(d)** Heatmap showing the PIF score distribution of 33 variants that show functional PIF in DMSO but are sensitive to either or both the DNA

damaging drugs.
(TIF)

**S6 Fig. Saturation genome editing of all possible SNVs from Exon 13. (a)** Pearson correlation showing the read abundance for each variant at day 3 between two independent replicates of Exon 13, ρ = 0.99 for 252 SNVs of Exon 13. **(b)** Percentage of HDR as quantified by the proportion of SNVs containing the PAM modifications. **(c)** Heatmap showing the distribution of indels formed at each exon. The frequency is normalized to the total read counts and z-score for each base pair indels is plotted. Each lane corresponds to the two independent replicates. **(d)** A strong correlation of the function scores between two independent replicates for 252 SNVs for DMSO vs cisplatin (R = 0.86), DMSO vs olaparib (R = 0.81) and cisplatin vs olaparib (R = 0.89). Red dots represent nonfunctional, blue dot represents functional and gray dot represents intermediate category of variants. **(e)** Histogram showing the distribution of function scores of 252 SNVs of which 8 SNVs are known to be non-functional as it encodes for a stop codon, 43 variants are known to be functional as it does not change the amino acid. Remaining 201 variants were experimental and their function scores were bimodally distributed. **(f)** Histogram showing the categorization of 252 variants into functional (188 SNvs), intermediate (4 SNVs) and non-functional (60 SNVs). **(g)** The distribution of PIF calculated based on the distribution of function scores revealed a clear delineation between functional and non-functional SNVs and only 4 variants were in the intermediate zone. **(h)** ROC plot showing the sensitivity and specificity of classifying the ClinVar-reported variants from exon 13. (AUC = 0.967) **(i)** Negative correlation between SGE-derived function scores of 252 SNVs with Bayes-del score (ρ = -0.23) and CADD scores (ρ = -0.32). Blue dots represent the functional, cyan dots are intermediate and red dots are non-functional variants according to our PIF-based calculation.
(TIF)

**S1 File. List of oligos.**
(XLSX)

**S2 File. Detailed summary of the function score of single nucleotide variants across 11 codons and exon 13 along with PIF classification, computational predictions and ClinVar classification.**
(XLSX)

**S3 File. Detailed summary of the function score of entire datasets along with PIF classification and corresponding confidence interval (CI) values.**
(XLSX)

**S4 File. Scores from SpliceAI predictions for 11 codons across exon 3, 18 and 19 and exon13 of BRCA2.**
(XLSX)

## Acknowledgments

We thank members of our laboratory for helpful discussions and suggestions. We acknowledge the support from Jeff Carrel and Megan Karwan from the CCR flow cytometry core; Dr. Elizabeth Conner from the CCR Genomic core; Bao Tran and Jyoti Shetty from the CCR sequencing facility for library preparation and NGS. All figures in this manuscript are prepared using Adobe Illustrator v6 and the illustrations are made using a paid subscription to BioRender.com. The content of this publication does not necessarily reflect the views or

policies of the U.S. Department of Health and Human Services, nor does mention of trade names, commercial products, or organizations imply endorsement by the U.S. government.

## Author Contributions

**Conceptualization:** Sounak Sahu, Kajal Biswas, Shyam K. Sharan.

**Data curation:** Sounak Sahu, Teresa L. Sullivan, Alexander Y. Mitrophanov, Mélissa Galloux, Darryl Nousome, Eileen Southon, Dylan Caylor, Arun Prakash Mishra, Sandra Burkett, Kajal Biswas.

**Formal analysis:** Sounak Sahu, Alexander Y. Mitrophanov, Mélissa Galloux, Darryl Nousome, Tyler Malys, Shyam K. Sharan.

**Funding acquisition:** Shyam K. Sharan.

**Investigation:** Sounak Sahu, Shyam K. Sharan.

**Methodology:** Sounak Sahu, Teresa L. Sullivan, Alexander Y. Mitrophanov, Mélissa Galloux, Eileen Southon, Dylan Caylor, Arun Prakash Mishra, Christine N. Evans, Michelle E. Clapp, Sandra Burkett, Tyler Malys, Raj Chari, Kajal Biswas.

**Resources:** Sounak Sahu, Christine N. Evans, Michelle E. Clapp, Raj Chari.

**Software:** Alexander Y. Mitrophanov, Mélissa Galloux, Darryl Nousome.

**Supervision:** Tyler Malys, Shyam K. Sharan.

**Validation:** Sounak Sahu, Alexander Y. Mitrophanov, Mélissa Galloux.

**Visualization:** Sounak Sahu.

**Writing – original draft:** Sounak Sahu, Shyam K. Sharan.

**Writing – review & editing:** Teresa L. Sullivan, Alexander Y. Mitrophanov, Mélissa Galloux, Darryl Nousome, Eileen Southon, Dylan Caylor, Arun Prakash Mishra, Christine N. Evans, Michelle E. Clapp, Sandra Burkett, Tyler Malys, Raj Chari, Kajal Biswas.

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
