## [Decision Letter · Decision Letter 0]

8 May 2023

Dear Dr Sharan,

Thank you very much for submitting your Research Article entitled 'Saturation Genome Editing coupled with chemotherapeutic drug response accurately determines pathogenicity of BRCA2 variants' to PLOS Genetics.

The manuscript was fully evaluated at the editorial level and by independent peer reviewers. The reviewers appreciated the attention to an important problem, but raised some substantial concerns about the current manuscript. Based on the reviews, we will not be able to accept this version of the manuscript, but we would be willing to review a much-revised version. We cannot, of course, promise publication at that time.

If you decide to revise the manuscript for further consideration at PLOS Genetics, please aim to resubmit within the next 60 days, unless it will take extra time to address the concerns of the reviewers, in which case we would appreciate an expected resubmission date by email to plosgenetics@plos.org.

We are sorry that we cannot be more positive about your manuscript at this stage. Please do not hesitate to contact us if you have any concerns or questions.

Yours sincerely,

Douglas M Fowler

Guest Editor

PLOS Genetics

Scott Williams

Section Editor

PLOS Genetics

Reviewer's Responses to Questions

**Comments to the Authors:**

Reviewer #1: In this paper the authors perform saturation genome editing of a region of BRCA2, an important breast and ovarian cancer gene. The manuscript presents a novel humanised-mouse ES cell model which is functionally validated - the text and figures are generally clear. I have several specific comments on this paper.

1. Throughout the manuscript the authors call VUS "variants of unknown significance". ACMG calls VUS, variants of uncertain significance. In a similar vein the authors state that "Large-scale genetic sequencing has not been widely used as a clinical paradigm due to our limited ability to determine the functional impact of genetic variants. " - I would argue this is far from the situation with genetic testing programmes expanding exponentially. I also think that the use of "functional" and "non-functional" variant in the paper might confuse some people. Maybe "disruptive" and "non-disruptive" is better.

2. The paper describes editing of just 11 codons. I was unable to find a clear description of why the authors selected these 11 codons and feel this is important for a general genetics journals such as Plos Genetics. Of note not all possible changes were analysed and there is some missing data making this a relatively small dataset. Only one gRNA was used.

3. In looking at Figure 2/3 it is clear that there is not a big delineation between "functional" and "non-functional" (disruptive and non-disruptive) variants. To what extent does this influence the utility of these method for analysis of the whole gene and indeed have the authors considered using different models to try and resolve more of the intermediate variants?

In sum this is an interesting model system for exploring BRCA2 albeit described here with a very small dataset.

Reviewer #2: Sahu and colleagues report a pilot-scale saturation genome editing scan of the tumor suppressor gene BRCA2. They develop a mESC model bearing a single copy of human BRCA2 on a BAC transgene, which they then subject to multiplex CRISPR/Cas9-medaited editing to knock in “NNN” codon libraries across 11 selected codons. Deep sequencing is used to measure the resulting allelic fractions at two timepoints as well as after drug treatment, which then contribute to a score taken as an estimate of the likelihood of pathogenicity.

BRCA2 contributes a heavy load of clinical VUS, so scalable approaches to generate function data in its variants are needed. This manuscript takes a modest step in that direction, though it leaves out important details and analyses as I describe below.

Major issues

- Scale. The experimental system shows promise as a platform for BRCA2 VUS functional profiling. However, targeting only 11 codons does leave open the question of whether it will be feasible to scale to all >3000 codons cDNA or to the undefined number of codons which contribute to the VUS burden.

- Drug selections. Function scores from the drug treatments (cisplatin, olaparib) are very closely correlated with function scores from outgrowth in DMSO (comparably so to DMSO-DMSO replicate correlations). It was not clear whether these selections added any information beyond that of additional replicates. Did any individual alleles behave qualitatively differently under either/both of those selections vs in DMSO?

- Sequencing data processing.

o Basic QC characteristics of the sequencing data were not shown. E.g., what fraction editing was obtained? What level of background mutations arise purely from sequencing error (sequencing WT cells or no-Cas9 transfection)?

o Different exons’ function scores (Fig S2b) and editing frequencies (Fig S2c) seem to span very different ranges. Are there any variants (e.g., synonymous SNVs) that could be used to normalize the exons to one another?

o Figure 2c. is there a single cutoff that perfectly separates the nonsense and synonymous variants? This is a small enough number of points (n=21 and 29) that they should be shown as individual measurements (e.g., in a swarm plot) instead of using binning, which obscures details.

- Exon 3 effects. As seen in Fig 2b and S2d, frameshift indels in exon 3 seem to persist without fitness defect between day 3 and 14. What might explain this apparently prevalent false negative? Does this affect interpretation of other effects (or lack of) in that exon?

- Validation experiments.

o Why are these particular missense variants chosen (beyond being MNVs and therefore not previously well studied)?

o In Fig 4A, WT curves seem different between the bottom and top – do these represent different experiments? No synonymous control is shown for exon 18, and p.Leu2768Lys is mentioned in the text but does not appear in the figure

- Splicing effects are mentioned several times but there are no results nor analyses to support these. How sure can we be that the up to 3 bp changes introduced, plus the PAM-ablating mutation, do not affect splicing? A though comparison should be made between any previous evidence for/against splicing disruption and the measurements made here. In addition, SpliceAI or similar should be used to score the sequence level changes made here to see how many of these effects could be attributed to splicing.

- Comparison to other high-throughput functional assays. How well correlated are these results to those of Ikegami (ref 21), Hanna et al (ref 27), and Cuella-Martin et al (ref 28)?

- Written clarity. The text ends abruptly without discussion which could help place these results in the context of other functional studies or additional lines of evidence. The writing is at times hard to parse or unclear.

- Variant nomenclature. Protein level variants are referred to extensively in the text, which made it unwieldy to connect these to cDNA-numbered variants e.g., shown in Figure 3e. The text should mention the “c.” numbers for each variant as well as protein level change.

Minor

- Fig 1b, what is being probed?

- In 1c, what is the expected copy number and what conclusions are drawn from the Tg measurements being > those of TK?

- is the antibody specific to human BRCA2 such that this blot shows only the human gene’s expression? If the case, this does not demonstrate equivalent expression between the transgene and endogenous copies.

- Fig 3a. PIF scores as previously shown are probabilities. In 3/3b, they are shown ranging from -1 to 2. Specify how they were transformed.

- Fig 3. Why the switch from PIFs to single-assay log2 scores?

Reviewer #3: Specific comments

The title is misleading – implies a large-scale MAVE style study – the study is impressive compared to previous BRCA2 functional assays but at the same time covers a limited number of codons (11) in only 3 exons. The title should be adapted to justify why analysis of just the 11 codons selected was considered most relevant – this is not in the abstract either. There is mention of why much later in the m/s on page 11 but there is no indication of what proportion of the functional domain/s of likely clinical importance have been targeted.

- Very first line of abstract is poorly phrased, should be reordered.

- It is unclear what the BRCA consortium in ref 20 refers to

- It is unclear why references 35-38 are referred to as clinical repositories – they are previously published functional assay studies?

- Assuming by the language that the southern blot was to BRCA2, not sure where in the text the detail for what the southern blot probe design is, or what it detects?

- Figure legend 1. “Neomycin resistant clones were

o screened for the presence of BRCA2.” Unclear from the wording of this if neomycin was used to select clones, and validated for BRCA1 or if the mESC electoporated ells were selected by neomycin and then BRCA2 presence validated, if so, how was this validated? Please clarify this?

o What is F7 and F11, it is no tin the legend, it is in the text but use consistency for clarity.

- Figure 2

o Fig 2c What is the possible explanation for the bimodal distribution in scores for stop codon variants?

o It is stated that the synonymous variants have a positive score - perhaps there is misrepresentation of the numbering on the x axis, but it appears from the figure as if at least some of the synonymous variants have a negative score?

o 2d sloppy terminology – the data represented should be referred to as a measure of function not reflective of pathogenicity. There is no clinical classification represented that subfigure.

o 2e. Why are there no red or blue CI lines, and these are not referenced in the figure legend to clarify, assuming the three colours are the ‘vertical lines’? More explanation is needed to describe the various categories experimental etc etc. The term classification should not be used in this context, to avoid confusion with the final clinical classification (eg ACMG) for a given variant. Suggest categorization perhaps. The comment is made in the legend “Cis were of considerable size due to the model’s ability to perform sharp separation”. Can the authors explain this statement - CI for individual variants is a measure of confidence in the repeatability of results for a variant, surely. So do these results indicate that some variant repeats for the assay had readout of no function and others as functional??

o 2f legend – again, please avoid the term classification in this context to avoid confusion.

Pg 12 : the authors state “The results obtained from functional assays are not always binary due to the cell-line-specific differences, technical or experimental variability. Hence, the development of a statistical classifier to generate a function-impairment probability, such as PIF is useful for the integration of results from multiple functional assays, providing a more reliable assessment of the impact of VUS.” Not at all sure how relevant this is to this study since all assays were done in one line – can the authors explain what they mean? Is this actually referring to the three exposures used in the experimental design.

The comparison to bioinformatics methods is not very helpful. Bioinformatic prediction of protein impact is not reliable (which is why unless calibrated it is recommended to be used at only supporting evidence). As the authors state, similar comparisons have shown up discrepancies, so what value to do again?

A more meaningful comparison would be to previous functional assays for the subset of variants with such data available and/or comparison to classification assertions in clinvar (which they do to some extent later- a more comprehensive summary tabulated as supplementary information would be helpful and save a lot of text).

“Furthermore, our variant classification is consistent with other functional assays” – again the term classification should not be used for categorization based on functional assay results alone.

Figure 3 - the key has grey dots as intermediate but there are none in the figure.

Figure 3 a,b why are both needed? And where does the terminology for null etc come from

“p.Pro2767Ser is identified in a patient with a benign cancer.” No cancer type or reference is provided.

“Exon 3 is a critical domain for PALB2 binding and has been previously reported to include several variants affecting splicing (51).” It is not clear what relevance this statement has to the discussion at that point which appears to be around missense variation?

“Since BRCA2 variants that are non-functional and will not survive in Brca2ko/ko mESCs, we have focused on the variants that are functional based on our PIF classification” - remove the word “and”.

Please justify the rationale to designate p.Leu2510Pro as hypomorphic.

- Supp Figure 4d – explain what experimental means – assumedly the variants that are not truncating or synonymous? The intermediate variants are supposed to be in grey –not clear where they are in the figure. Can they be marked with a hash or something? Do the figures on the RHS include controls – apparently since the n=599 is the same as on the LHS.

- Please highlight the intermediate variants reported with PIF values in the figure – hard to place them in context with the other variants.

- Figure 4

o Why are the control cells at a different magnification to the variant expressing cells. Do you have an IR negative or a

- Methods section 7 – change functional classification to categorization – again to avoid confusion with ACMG terminology.

-

General comments

- The methods are throughout the main manuscript, methods and figure legends, and it would appear that details of the methods are missing, though it is hard to be sure with the way it has been written.

- The selection of 1 single clone to represent a humanised mESC line is questionable. Would like the authors to comment on the similarity or difference in behaviour of the single clone, relative to the parental line. Doubling time, behaviour etc.

- Training set of variant is “21 functional synonymous variants with no change in amino acid and 29 non-functional variants generating a stop codon”. Using loss of function variants as a predictor of missense variant pathogenicity might be questioned, given the different possible mechanisms by which a missense variant might impact the protein function. It would be helpful to have existing known pathogenic and benign variants designated, and used for a formal calculation of Likelihood ratio towards pathogenicity as recommended by Brnich et al.

- It’s unclear from any of the text if there are any biological replicates. “two independent replicates” were used, but these appeared to be performed concurrently. The biological variation in variant reads is not clear, the authors should clarify if the two independent replicates were performed in completely independent experiments and how biologically variant these were (though the use of the same clone as ‘independent’ is also questioned). This also might be supported by combining the figures in SUPP 2c, though duplicates does not really suffice for showing variability in an assay.

- Can the authors comment on the variability of reps for the variants called intermediate?

Overall the paper is not easy to read and figures need more explanation in the figures themselves and the legends.

Formal comparison to previous larger-scale studies is essential as is calculation of LR towards pathogenicity using ACMG recommendations – particularly since the authors stress many times how important functional information is for classification.

It would be helpful also for the authors to state explicitly if/how the assay captures potential effects on splicing and which variants in their assay might be considered to be deleterious due to this mechanism of impact.

**Have all data underlying the figures and results presented in the manuscript been provided?**

Reviewer #1: Yes

Reviewer #2: Yes

Reviewer #3: Yes

PLOS authors have the option to publish the peer review history of their article (what does this mean?). If published, this will include your full peer review and any attached files.

Reviewer #1: No

Reviewer #2: No

Reviewer #3: No

---

## [Decision Letter · Decision Letter 1]

14 Aug 2023

Dear Dr Sharan,

Thank you very much for submitting your Research Article entitled 'Saturation genome editing of 11 codons and exon 13 of BRCA2 coupled with chemotherapeutic drug response accurately determines pathogenicity of variants' to PLOS Genetics.

The manuscript was fully evaluated at the editorial level and by independent peer reviewers. The reviewers appreciated work you have done to revise the initial submission but identified some concerns that we ask you address in a revised manuscript.

We therefore ask you to modify the manuscript according to the review recommendations. Your revisions should address the specific points made by each reviewer.

Yours sincerely,

Douglas M Fowler

Guest Editor

PLOS Genetics

Scott Williams

Section Editor

PLOS Genetics

Reviewer's Responses to Questions

**Comments to the Authors:**

Reviewer #2: This revised manuscript from Sahu and colleagues is improved and addresses most of my concerns. In particular, adding the full exon 13 SGE goes a long way towards demonstrating the scalability of the approach. To preview the road ahead, the authors should also include a rough estimate as to the number of experiments required to flesh out the whole gene, or at least however much of it they feel has sufficient pathogenic missense burden to justify the work.

My major remaining concern is around the claims that the assay can identify partial loss of function mutations using DNA damaging agents. ( “BRCA2 variants with a loss of function are expected to be depleted … and can be regarded as pathogenic, whereas variants with a partial loss of function may not be significantly depleted by day 14. However, these latter variants are expected to exhibit sensitivity to DNA-damaging drugs” )

This portion of the manuscript remains poorly supported. I do not see any evidence that the drug treatments are any adding information beyond that provided by the DMSO treatment, for the following reasons: (1) Nonsense-synonymous separation seems to be nearly identical between DMSO, cisplatin, and Olaparib. (2) In the plots provided, DMSO-DMSO replicate difference seems comparable to that of either compound versus the other or versus DMSO (3) That one-condition PIF models outperform the full model probably just reflect the fact that there are more replicates in the latter.

Moreover, the “NNN” codons provide an opportunity for internal replication by virtue of multiple codons encoding for the amino acid change; for the examples of intermediate effect the authors highlight, these point to the explanation being an averaging of noisy data rather than a true intermediate/hypomorphic effect. For instance, an example is given of p.Glu58Val (c.173_174AA>TG), which in Table S3 is listed as having a final PIP score of 0.278 (note—not less than 0.2 as the manuscript states). The other three equivalent codon swaps (GAA>GTA,GTC, and GTT) all have PIP scores of ~0, and are classified as functional.

I think the manuscript would be stronger without the claims about intermediate variants, but if included, they should be better supported.

Minor

- Figure legends are really on the long side

- “Humanized HEK293T cells” should be “Haploidized HEK293T cells” (it is a human cell line)

- Terminology throughout is still confusing and a bit nonstandard. In several places “non-functional” and “functional” are used to mean “nonsense” and “synonymous”, which as the more specific and descriptive terms should be used. “Experimental” seems to be used to refer to all other elements of the library, which is similarly vague; how about “library” or “missense library” for that?

“We found a strong correlation… between the two independent replicates across exon 3, 18, and 9” (should 9 be 19 here?)

Reviewer #3: The authors have done considerable work to address a number of comments, and the manuscript is substantially improved. In addition, expansion of the assay across exon 13 is a considerable improvement to show that this is a significant approach.

However. there are still some points that need addressing, some relating to the original questions, and some due to additional edits.

The grammar in the first sentence of the abstract still needs work. Two grammatical errors remain.

The new distribution plots in Fig 2c are in improvement, and do show that there is overlap in the scores for synonymous and stop codon variants, albeit minimal. Could the authors include an F test to confirm normal distribution. And would it be possible to generate a plot similar to Fig 2c where the PIF based on all three assays is displayed only for the synonymous and stop codon variants, in order to show separation of PIF for these variant as determined based on the three scores combined.

Fig 2d – thank-you for the correction. Could the fig 2 title also be amended eg change “to classify” to “to inform classification of”

The authors have added supplementary file 2 but appear to have not even referred to it in the amended text. More discussion is required, including a discussion of how variants/results overlap with what was included in training dataset in addition to the synonymous and stop codon variants.

Additional comments relating to calibration and clinical evidence strength calculation.

The new manuscript page 30 (page 59 of the document), methods section 9, states “We calculated positive Likelyhood Ratio (LR+) and negative Likelyhood Ratio (LR-) to provide an indication of the probability of a variant being pathogenic based on our functional assay results.”

First - likelyhood ratio should read “likelihood ratio”.

Regarding the calculation for positive LR and negative LR – as written the calculations seem to assume that there is no grey zone – no region where the functional impact is unclear. Based on results shown in fig 2c, there would be value in assigning a zone of functional score as “unclear”, where both synonymous and stop codon variants overlap in functional score.

Could the authors confirm if only the synonymous and stop codon variants were used to define this sensitivity and specificity? Or were the clinvar LP and PB variants used as well (see below). If so then those variants should also be plotted in Fig 2c and any plot that shows the PIF based on all three scores.

The authors state that the Brnich calibration method cannot be used, as this requires multiple lines of evidence. There appears to be some confusion since the authors could use previously classified variants (eg those in clinvar) as controls to calibrate. Indeed they might have done this, as per this text in the methods: “ All non-sense variants and all variants classified as “Pathogenic” or “Likely pathogenic” in ClinVar database constituted the “Non-functional” class in the training dataset. All synonymous and all variants classified as “Benign” or “Likely benign” in ClinVar database constituted the “Functional” class. 42 known functional and 12 known non-functional variants were used for the training dataset.”

Could the authors please clarify where these numbers come from, and how they do or do not overlap with the numbers in figure 2c (21 synonymous and 29 stop codon).

A reasonable approach would be to use the synonymous and stop codon variants to set the original cutpoints for what is functional, not functional and intermediate (where scores are overlapping, and then formally calibrate the functional assay categories in an independent set of variants of known pathogenicity from ClinVar. Obviously it would be important to assess if these variants could be classified without use of other functional data, but it should be possible. That would add value for interpretation of findings since the application of the Gaussian mixture model is not inherently intuitive to curators, and would allow simple application of LR cutoffs determined for weighting ACMG codes. Given that the authors are stressing the potential clinical importance of their study, doing the additional legwork to calibrate using the Brnich* approach would seem to be in scope and a lot easier than doing extra functional assays. That is, please calculate the diagnostic odds of pathogenicity for the assay based on a reference/calibration set that does not overlap with the original set of variants use to set the thresholds.

*Brnich, S.E., Abou Tayoun, A.N., Couch, F.J. et al. Recommendations for application of the functional evidence PS3/BS3 criterion using the ACMG/AMP sequence variant interpretation framework. Genome Med 12, 3 (2020). https://doi.org/10.1186/s13073-019-0690-2.

Splicing - The manuscript has added in information regarding the consideration of splice variants. Noting that the SpliceAI results have been added to Supp File 4, it is not clear how these SpliceAI results were used to inform potential impact on splicing. Specifically, the authors state “Furthermore, sgRNAs were selected based on SpliceAI predictions such that the synonymous PAM mutation does not affect canonical splicing”. It appears in the table that there are SpliceAI scores well above 0.2. Could the authors please clarify exactly what cutpoints they used to assess potential impact on splicing. A recent calibration experiment arising from a ClinGen group (https://doi.org/10.1016/j.ajhg.2023.06.002) recommends conservative cutpoint of 0.2 to indicate potential splicing impact. The data provided in Supp File 4, at least at a quick glance, would suggest that some of the variants have potential to alter splicing… If all of these were included as PAM alterations, then this this could have implications for interpretation of the functional assay findings

Additional comment. There still appears to be ambiguity in the difference between pathogenic (as in a validated, curated, clinical disease associated variant) and one that scores as loss of function as per assay results from this study.

**Have all data underlying the figures and results presented in the manuscript been provided?**

Reviewer #2: Yes

Reviewer #3: Yes

PLOS authors have the option to publish the peer review history of their article (what does this mean?). If published, this will include your full peer review and any attached files.

Reviewer #2: No

Reviewer #3: No

---

## [Editor Report · Decision Letter 2]

28 Aug 2023

Dear Dr Sharan,

We are pleased to inform you that your manuscript entitled "Saturation genome editing of 11 codons and exon 13 of BRCA2 coupled with chemotherapeutic drug response accurately determines pathogenicity of variants" has been editorially accepted for publication in PLOS Genetics. Congratulations!

Yours sincerely,

Douglas M Fowler

Guest Editor

PLOS Genetics

Scott Williams

Section Editor

PLOS Genetics

Comments from the reviewers (if applicable):

**Data Deposition**

http://datadryad.org/submit?journalID=pgenetics&manu=PGENETICS-D-23-00242R2

**Press Queries**

---

## [Editor Report · Acceptance letter]

13 Sep 2023

PGENETICS-D-23-00242R2 

Saturation genome editing of 11 codons and exon 13 of *BRCA2* coupled with chemotherapeutic drug response accurately determines pathogenicity of variants 

Dear Dr Sharan, 

We are pleased to inform you that your manuscript entitled "Saturation genome editing of 11 codons and exon 13 of *BRCA2* coupled with chemotherapeutic drug response accurately determines pathogenicity of variants" has been formally accepted for publication in PLOS Genetics! Your manuscript is now with our production department and you will be notified of the publication date in due course.

With kind regards,

Jazmin Toth

PLOS Genetics

On behalf of:
